# Conformational transitions and allosteric modulation in a heteromeric glycine receptor

Eric Gibbs[1], Emily Klemm[1], David Seiferth [2,6], Arvind Kumar [1,6], Serban L. Ilca [3,4], Philip C. Biggin [2] & Sudha Chakrapani [1,5] ✉

Glycine Receptors (GlyRs) provide inhibitory neuronal input in the spinal cord and brainstem, which is critical for muscle coordination and sensory perception. Synaptic GlyRs are a heteromeric assembly of α and β subunits. Here we present cryo-EM structures of full-length zebrafish α1β_BGlyR in the presence of an antagonist (strychnine), agonist (glycine), or agonist with a positive allosteric modulator (glycine/ivermectin). Each structure shows a distinct pore conformation with varying degrees of asymmetry. Molecular dynamic simulations found the structures were in a closed (strychnine) and desensitized states (glycine and glycine/ivermectin). Ivermectin binds at all five interfaces, but in a distinct binding pose at the β-α interface. Subunit-specific features were sufficient to solve structures without a fiduciary marker and to confirm the 4α:1β stoichiometry recently observed. We also report features of the extracellular and intracellular domains. Together, our results show distinct compositional and conformational properties of α_1βGlyR and provide a framework for further study of this physiologically important channel.

Glycine receptors (GlyRs) are chloride-conducting, pentameric ligand-gated ion channels (pLGIC) primarily found in spinal cord neurons where they mediate fast inhibitory neurotransmission[1,2]. Their inhibitory input is essential for coordinated muscle movement, as manifested in hyperekplexia[3], a neurological disorder where mutations affecting GlyR clustering or function result in sporadic muscle movements. GlyRs also play an important modulatory role in visual, auditory, and pain perception[4–6]. As such, GlyR-positive allosteric modulators (PAMs) have promise as therapeutics for chronic and inflammatory pain[7]. Structurally, GlyRs assemble as pentamers composed of homologous α and β subunits. There are four genes encoding α subunits (α1–α4), each capable of forming functional homomeric channels. However, synaptic localization requires the β subunit as it solely interacts with gephyrin, a synaptic-scaffold protein[8]. The majority of adult GlyRs are composed of α_1 and β subunits[9,10]. Recent studies have reported structures of heteromeric GlyR expressed in HEK cells (α_2β) and from native porcine brain tissue (α_1β)[11,12]. These structures agreed with the surprising finding that heteromeric GlyR has a 4α:1β stoichiometry, contrary to prior predictions from biochemical analyses, which indicated a 2α:3β or 3α:2β stoichiometry[13–18]. However, functional interpretation was limited by the use of protein alterations or antibodies, which aid in cryo-EM image processing but potentially alter native channel properties[19].

Here we report three structures of full-length zebrafish (ZF) heteromeric GlyR composed of α_1 and β_B subunits absent of any exogenous fiduciary markers. In each, we confirm the 4α:1β stoichiometry by identifying subunit-specific features. These include previously uncharacterized portions of the β subunit in the extracellular domain (ECD) and intracellular domain (ICD) that have important implications for subunit assembly and channel function. Structures from data in the

[1]Department of Physiology and Biophysics, Case Western Reserve University, Cleveland, OH 44106-4970, USA. [2]Department of Biochemistry, University of Oxford, Oxford OX1 3QU, UK. [3]New York Structural Biology Center, New York, NY 10027, USA. [4]Simons Electron Microscopy Center, New York, NY 10027, USA. [5]Department of Neuroscience, School of Medicine, Case Western Reserve University, Cleveland, OH 44106-4970, USA. [6]These authors contributed equally: David Seiferth, Arvind Kumar. ✉e-mail: Sudha.chakrapani@case.edu

presence of 100 µM strychnine (αβGlyR-Stry at 2.97 Å) and 1 mM glycine (αβGlyR-Gly at 3.21 Å) were in a closed and desensitized conformation, respectively. These confirmations are distinct from those observed in truncated channels, but consistent with those observed in native channels[11] and full-length homomeric αGlyR structures[12,20–23]. To further explore conformational heterogeneity within αβGlyR, an additional structure was solved in the presence of 1 mM glycine and 20 µM ivermectin (αβGlyR-Gly-Ivm at 2.98 Å). Ivermectin is a GlyR partial agonist/PAM whose binding site is wedged between the M3 helix on the principal subunit and the M1 helix on the complementary subunit[20,21,24–28]. In homomeric αGlyR, mutation of a conserved M3 alanine (A312 in ZF-α$_1$GlyR) to a larger side chain ablates ivermectin activity and potentiation. As βGlyR has isoleucine at this position (I333), it is generally thought that ivermectin may not bind at the β/α interface or binds in a way that does not promote channel activity[26]. Unexpectedly, our structure shows clear ivermectin densities at all five subunit interfaces. However, the molecule adopts a distinct pose at the β/α interface because of the larger side chain volume at I333. The α/β interface also has a distinct interaction profile due to side chain differences between α and β subunits. αβGlyR-Gly-Ivm has a slightly different pore conformation relative to αβGlyR-Gly. Specifically, it is more constricted at the intracellular end of the pore-lining M2 helix. Molecular dynamics (MD) simulations confirm the channel is in a non-conducting, desensitized state. Together, the three αβGlyR structures build on recent past work to deepen our understanding of how subunit heterogeneity impacts the structure and function of heteromeric GlyR.

## Results

### Functional validation and cryo-EM maps of ZF-αβGlyR

The Zebrafish (ZF) genome has five α genes (α$_1$–α$_3$, α$_{4a}$, and α$_{4b}$) and two β genes (β$_z$ and β$_B$), with β$_B$GlyR having a slightly higher sequence identity to the sole human βGlyR gene (85% vs. 81%). Similar to human (h) GlyR genes, the α subunits express as functional homomeric channels while the β subunits do not[18]. ZF-GlyRs and h-GlyRs are pharmacologically similar, with a notable distinction that ZF-αβGlyR is blocked by low µM amounts of picrotoxin, whereas the h-αβGlyR is insensitive to picrotoxin[18]. This is largely due to a species-specific difference in the 6′ position on the pore-lining M2 helix, a leucine in both ZF-βGlyR genes, and phenylalanine in h-βGlyR[29]. Previous work has shown that ZF-αβGlyRs are less sensitive to glycine than ZF-αGlyRs when expressed in *Xenopus* oocytes, similar to the differences seen between h-αβGlyR and h-αGlyR[14,18,30]. The functionality of the full-length ZF constructs, ZF-α$_1$GlyR (referred to as αGlyR hereafter) and β$_B$GlyR (referred to as βGlyR hereafter), used in the cryo-EM studies were tested in HEK 293 T cells. α$_1$GlyR homomers or α$_1$β$_B$GlyR heteromers display glycine-induced currents with an EC$_{50}$ of $19 \pm 11$ µM and $63 \pm 13$ µM, respectively (Fig. 1A, B; Supplementary Tables 1 and 2). A right shift in a glycine dose response for heteromeric αβGlyR is consistent with measurements in both human and zebrafish genes[18,31].

Baculovirus for αGlyR and βGlyR was generated using the pFastBacDual plasmid, subcloned with codon-optimized full-length GlyR genes (Supplementary Table 3)[21]. Baculovirus for the GlyR-binding domain of rat gephyrin (Geph-E) was generated using the pFastBac1 plasmid, subcloned with codon-optimized Geph-E gene (Supplementary Table 4). Protein was expressed in Sf9 cells co-infected with the dual baculovirus for αGlyR/βGlyR, and separate baculovirus for Geph-E in a 0.8:1 ratio. α$_1$β$_B$GlyR (αβGlyR) heteromers were purified using two affinity purification steps, one for each subunit, to eliminate homomeric assemblies. After gel filtration, the protein eluted as a single peak, and western analysis showed bands corresponding to all three genes (Fig. 1C). The elution volume is 0.7 ml left-shifted compared to homomeric α1-GlyR and α1βGlyR (without Geph-E) preparations under similar conditions. The Mass Photometry analysis of the main gel filtration peak of αβGlyR-Geph-E gives a molecular mass estimate of 396 kDa, which is close to the predicted mass of 405.5 kDa

(Supplementary Fig. 1)[32]. A homomeric assembly of αGlyR would be expected to have a molecular weight of 293.1 kDa. Hence the data reasonably exclude a sizable population of homomeric αGlyR. The αβGlyR-Geph-E samples were vitrified on cryo-EM grids in the presence of either 100 µM strychnine, 1 mM glycine, or 1 mM glycine and 20 µM ivermectin. The vitrified samples were imaged on a Titan Krios 300 keV electron microscope, and single-particle analysis was carried out for each of these conditions (Supplementary Fig. 2). The particles from each sample condition were refined to one conformational state with three-dimensional reconstruction at a nominal resolution of 2.97 Å (αβGlyR-Stry), 3.21 Å (αβGlyR-Gly) and 2.98 Å (αβGlyR-Gly-Ivm) (Supplementary Figs. 3–5).

Similar to other GlyR structures[11,12,20–24,33], the 3D reconstructions in each conformational state included density for most of the ECD, TMD, and some parts of the ICD (Fig. 1D). The ICD is predicted to be largely unstructured and includes the gephyrin binding site[8]. Although Geph-E was not resolved in the final maps, the ICD region showed diffuse asymmetric features that may have been stabilized by the inclusion of Geph-E. These were best defined in the αβGlyR-Stry reconstruction. Similar to recent work with heteromeric nAChRs[34,35], there are several subunit-specific features observed at a moderate resolution that allowed for an unambiguous assignment of the 4α:1β stoichiometry, even in the absence of fiduciary markers (discussed later). The observed stoichiometry is consistent with two recent reports of heteromeric GlyR structures[11,12].

### Channel pore conformations

A comparison of αβGlyR-Stry, αβGlyR-Gly, and αβGlyR-Gly-Ivm conformations reveals distinct ion permeation pathways. The effects of subunit heterogeneity are particularly apparent at the level of M2 (Fig. 2) and are consistent with the different conductance and pharmacological properties of α$_1$βGlyR vs. α$_1$GlyR channels[14,31]. The conserved pore architecture of pLGICs has two channel gates. The first is the activation gate, which is located in the middle of the pore and formed by the highly conserved 9′ leucine (L9′)[36]. The second is the desensitization gate at the intracellular end of the pore, specifically the −2′ proline position in αGlyR[37] (P-2′). P-2′ constitutes the charge selectivity within a sequence conserved among αGlyR subunits (P-2′, A-1′, and A0′)[38,39]. However, in βGlyR, the −2′ position is an alanine (glycine in h-βGlyR), and proline is found at the 2′ position instead, which allows for a potential break in the helical structure of βGlyR near the desensitization gate. In the apo or orthosteric antagonist-bound state of homomeric αGlyR, L9′ side chains form a hydrophobic barrier at the activation gate[21,22]. Glycine-bound homomeric αGlyR is open at the activation gate but non-conducting at the αP-2′ gate[12,21,22,40]. Channel opening was observed as a symmetric process in homomeric GlyRs for both α$_1$ and α$_3$ channels, likely a result of averaging about the C5 axis. Asymmetric M2 movements have been observed in molecular dynamic simulations of homomeric α1GlyR[41]

In αβGlyR-Stry, the pore is mostly symmetric with the M2 helix nearly perpendicular to the membrane in all five subunits, creating a cylindrical pore (Fig. 2A). There are hydrophobic barriers both at L9′ (pore radius ~1.3 Å) and αP-2′ (βA-2′) (pore radius ~2.0 Å). Pore lining residues are poorly conserved between αGlyR and βGlyR and generally more hydrophobic in βGlyR. In particular, the 6′ position is a threonine in αGlyR but leucine in βGlyR (phenylalanine in h-βGlyR)[20,21,23]. The radius of the pore at 20′, 13′, 9′, 6′, and −2′ positions are below the Born radius for the solvated chloride ion, which is 2.26 Å (the Pauling radius for chloride ion is 1.81 Å)[42], and are therefore likely barriers to ion permeation (Fig. 2B). Notably, the pore at each of these positions is more constricted relative to closed homomeric αGlyR structures.

In αβGlyR-Gly, the pore is expanded along the permeation pathway at αT13′ (βS13′) and L9′, similar to the pore of glycine-bound homomeric αGlyR[21,22]. The L9′ side chains are rotated away from the

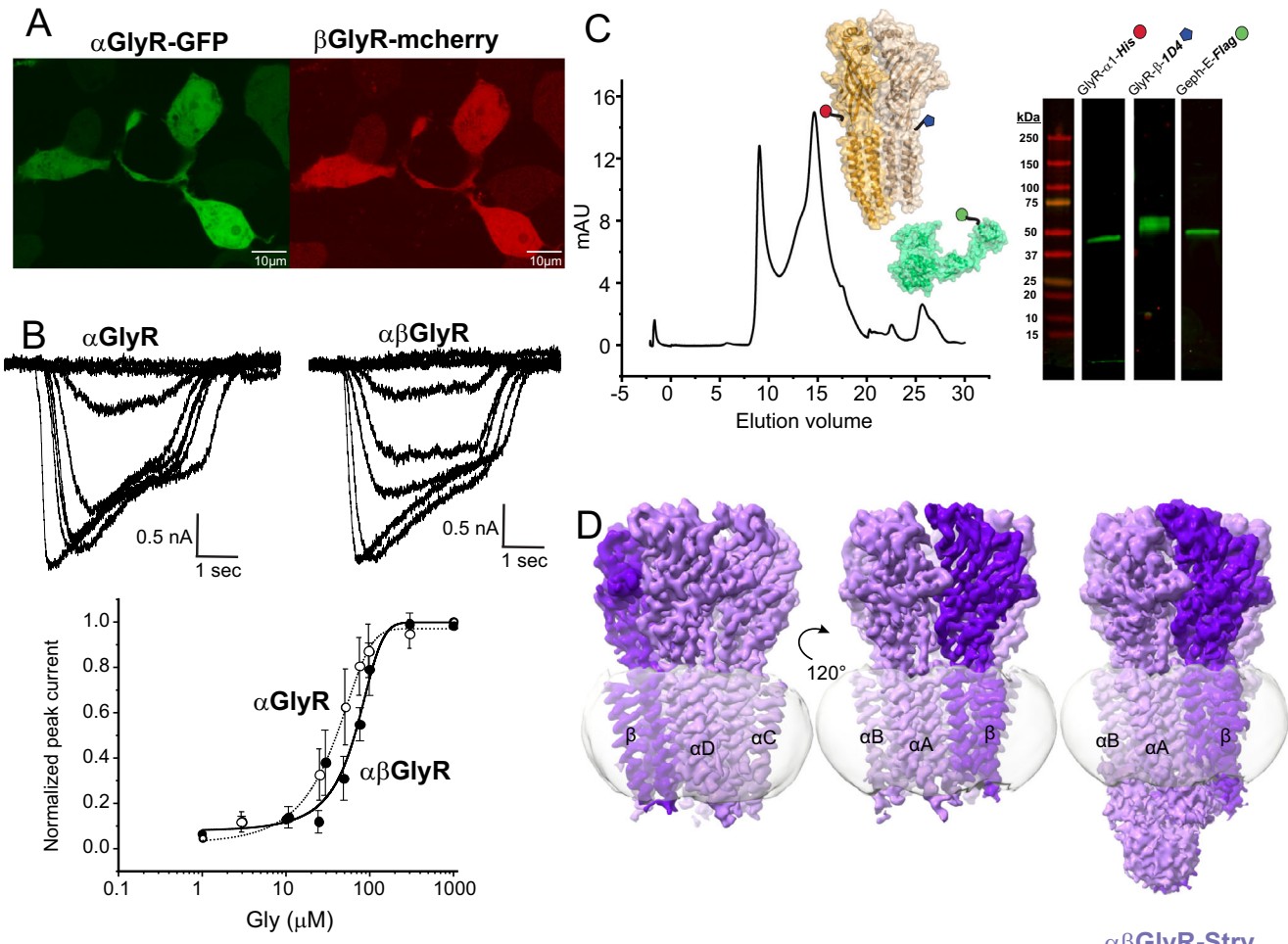

αβ**GlyR-Stry**

**Fig. 1 | Expression, purification, functional validation, and cryo-EM map of heteromeric αβGlyR. A** Confocal fluorescent microscopy images of HEK 293 T cells showing GFP and mCherry expression, which respectively co-expressed with αGlyR and βGlyR subunits, via a bicistronic plasmid. βGlyR was transfected at a 5:1 excess, even though the mCherry fluorescence appears weaker. Generally, transfection efficiency was around 50%, and similar fluorescent images were observed for all cell plates used for electrophysiology. **B** Whole-cell current recordings from HEK 293 T cells transfected with only αGlyR (left) and with αGlyR and βGlyR (right) in response to glycine (concentration range: 1–1000 μM). Dose–response curves were generated from recordings on different batches of cells. The $EC_{50}$ for αGlyR was $19 \pm 11\,\mu M$ ($n = 5$), and for αβGlyR was $63 \pm 13\,\mu M$

($n = 4$). Data are presented as mean values ± standard deviation. **C** αβGlyR (coexpressed with Geph-E) was purified with a two-step affinity purification using subunit-specific affinity tags for βGlyR (1D4) and αGlyR (8xHis) followed by separation using size exclusion chromatography. The main peak (~14.8 ml) was at the elution volume expected for a pentameric assembly, and western blot analysis showed the peak contained all three expressed proteins. Similar gel filtration profiles were consistent for about 20 purifications in the course of this study. **D** Final cryo-EM reconstructions of αβGlyR-Stry. The density corresponding to the beta subunit is colored in a darker shade of purple. The left and center figures are shown at $\sigma = 0.16$, and the rightmost figure is shown at $\sigma = 0.09$ to highlight the diffuse ICD signal.

pore and toward the neighboring M2 helix[41], with slight differences between subunits in their relative L9' positions (Fig. 2C). Although the pore is wider at αT6' (βL6'), the βL6' side chain still points towards the pore center, creating an asymmetric pore profile. This suggests that the analogous phenylalanine in human αβGlyR would also point to the center, and multiple βGlyR subunits could possibly result in a non-conducting pore as recent mammalian αβGlyR structures suggest[11,12]. Residues at the desensitization gate, αP-2' (βA-2'), are shifted up relative to their positions in αβGlyR-Stry, similar to what was observed in homomeric GlyR[21,22]. The desensitization gate is asymmetric at the P-2' position, with the αA interface radially outward from the pore center and slightly lower than the other −2' residues. (Fig. 2C, α subunits are labeled A–D, clockwise from the β subunit when viewed from the extracellular membrane). This asymmetry gives an elliptical pore profile at the desensitization gate, but the radius along the minor axis is 1.9 Å, similar to the radius observed in other desensitized heteromeric GlyR structures and past homomeric αGlyR structures[11,12,21,22]. The αβGlyR-Gly-Ivm pore conformation is similar to αβGlyR-Gly but

has a distinct desensitization gate. P-2' of αA is moved up and towards the pore axis. This creates a gate that is both more symmetric and constricted with a radius along the minor axis of 1.6 Å.

## Computational analysis of the channel pore in different conformational states

The conductance states for αβGlyR-Stry, αβGlyR-Gly, and αβGlyR-Gly-Ivm were assessed by MD simulations of each of these conformations embedded within a hydrated 1-palmitoyl-2-oleoyl-*sn*-glycero-3-phosphoethanolamine lipid bilayer. Computational analysis, similar to that done for homomeric αGlyR[21], was done to better understand the effects of heterogeneous subunit movement on channel conductance (Fig. 3). Simulations were carried out with positional restraints on the protein backbone to preserve the overall experimentally determined conformational state while permitting rotameric flexibility of the amino acid side chains. During the 50 ns of simulation runs, all three pore conformations remained stable with no major changes to the overall pore radii besides fluctuations arising from side-chain

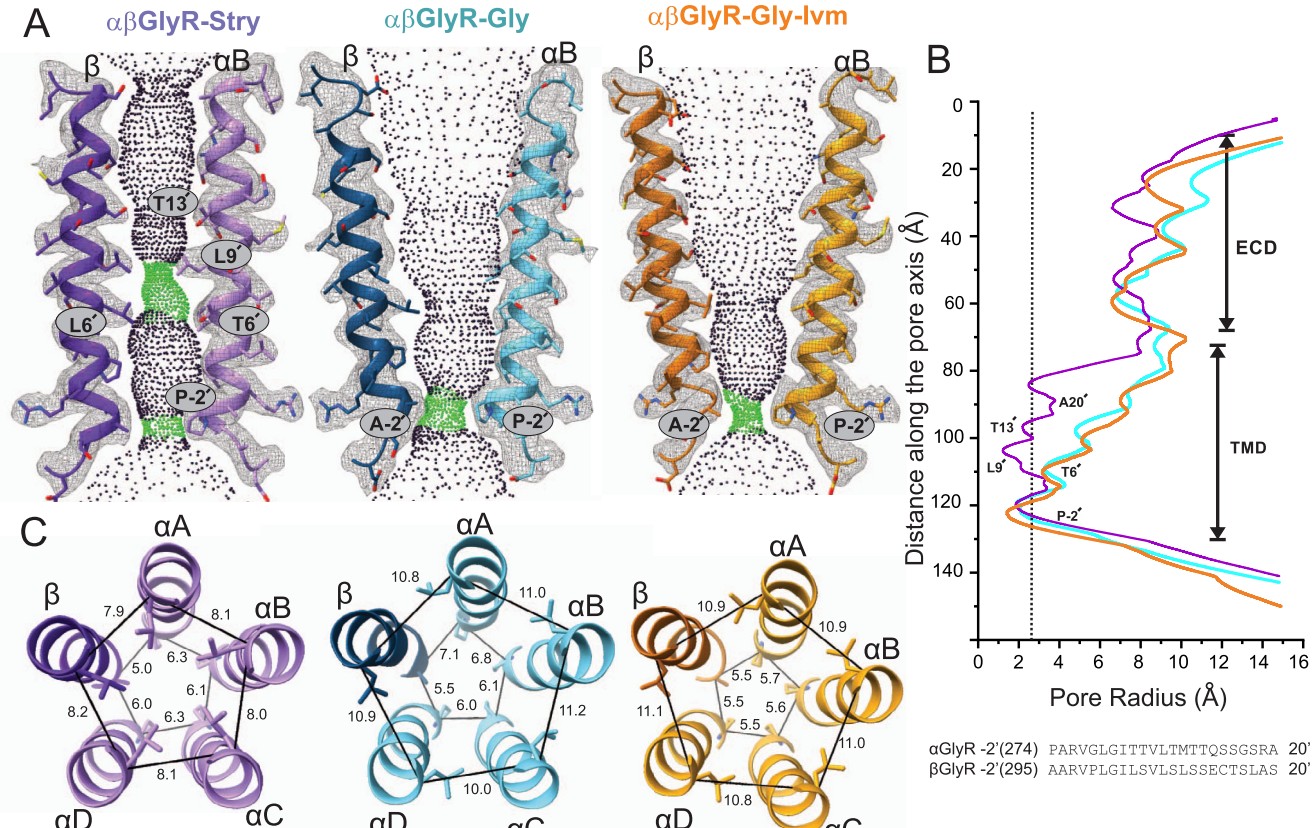

**Fig. 2 | Conformational changes within the αβGlyR pore in response to different ligands. A** Map and model representations of M2 helices in the αB and β subunits as seen in αβGly-Stry, αβGly-Gly and αβGly-Gly-Ivm (σ = 0.146, 0.25, and 0.13, respectively). The ion permeation profiles of the channel pore were generated using HOLE[85]. The side chains lining the constrictions (green dots) are labeled. **B** The smallest pore radii as a function of the position along the pore axis. The radius of a hydrated chloride ion is indicated as a dotted line and shows that all three states are expected to be non-conducting, though this analysis does not consider hydrophobicity or pore asymmetry. Residues lining the M2 helix for each subunit are shown below the plot. **C** Top-down views of the L9′ and P-2′ positions which respectively correspond to the channel activation and desensitization gates. Distance labels are in Å.

movements of residues lining the pore (Fig. 3A). pLGIC pores do not have well-defined ion coordination sites but rather pass solvated ions. As such, the pore hydrophobicity, as well as the pore radius, are determinants of channel conductivity[43,44]. MD simulations were done with flexible side chains to assess water density within each of the three pore conformations (Fig. 3B). αβGlyR-Stry showed clear channel dewetting near L9′ and occasional dewetting at αP-2′. In contrast, αβGlyR-Gly showed a hydrated pore at L9′ and partial dewetting of the pore near αP-2′ (βA-2′). αβGlyR-Gly-Ivm shows more dewetting than αβGlyR-Gly at αP-2′ (βA-2′), consistent with the smaller channel radius.

Two-hundred nanoseconds simulations were then carried out with backbone restraints in the presence of 150 mM NaCl and a transmembrane potential of +500 mV (i.e., positive at the cytoplasmic side) (Fig. 3C). Consistent with pore-wetting simulations, the αβGlyR-Stry simulation showed the area surrounding L9′ was dewetted through the entire simulation, and chloride ions were absent in the channel pore. The αβGlyR-Gly simulation showed water and chloride traversing the L9′ position. The αP-2′ (βA-2′) gate was mostly wetted throughout the simulation, but no chloride penetration was observed. As no ion permeation was observed, we conclude αβGlyR-Gly is in a desensitized state. The αβGlyR-Gly-Ivm simulation showed water and chloride penetration past the L9′ gate similar to αβGlyR-Gly. However, unlike αβGlyR-Gly, the area surrounding αP-2′ (βA-2′) was mostly dewetted throughout the simulation, and no chloride permeation was observed. We thus conclude that αβGlyR-Gly-Ivm is also in a desensitized conformation.

## Global conformational changes

To assess the global conformational changes associated with channel gating, the arrangement of subunits was compared across different functional states. The ECDs were mostly symmetric about the channel pore axis, and there were few changes in their relative positions between different states (Fig. 4A, Supplementary Table 6). In both αβGlyR-Stry and αβGlyR-Gly, the TMDs were mostly symmetric, but the TMD of αβGlyR-Gly subunits were displaced outward and rotated counterclockwise compared to those of αβGlyR-Stry (Note that the rotation of the TMD can only be measured relative to the ECD as there is no absolute reference.) The TMD of αβGlyR-Gly-Ivm subunits were arranged almost identically to αβGlyR-Gly. These changes are consistent with the twisting mode observed in structural and computational studies of other pLGICs, but the fixed center of mass within the ECD is distinct from the previously observed blooming[45–48]. This indicates there may be channel-specific variations to the general framework proposed for pLGIC activation.

Conformational changes within individual subunit domains were assessed by principal component analysis (PCA) of the same subunit in different conformations. The results separated conformational changes due to subunit rigid rotations and local deformations. Between αβGlyR-Stry and αβGlyR-Gly, each subunit ECD rotates by about 7° counterclockwise, when viewed extracellularly, and tilts towards the channel pore axis by about 4°. The subunit's TMD tilts towards the M3 helix by about 6° and rotates clockwise by about 11°. The exact values for each subunit are given in Supplementary Table 7. Figure 4B shows the difference between main chain atom positions in αβGlyR-Stry and

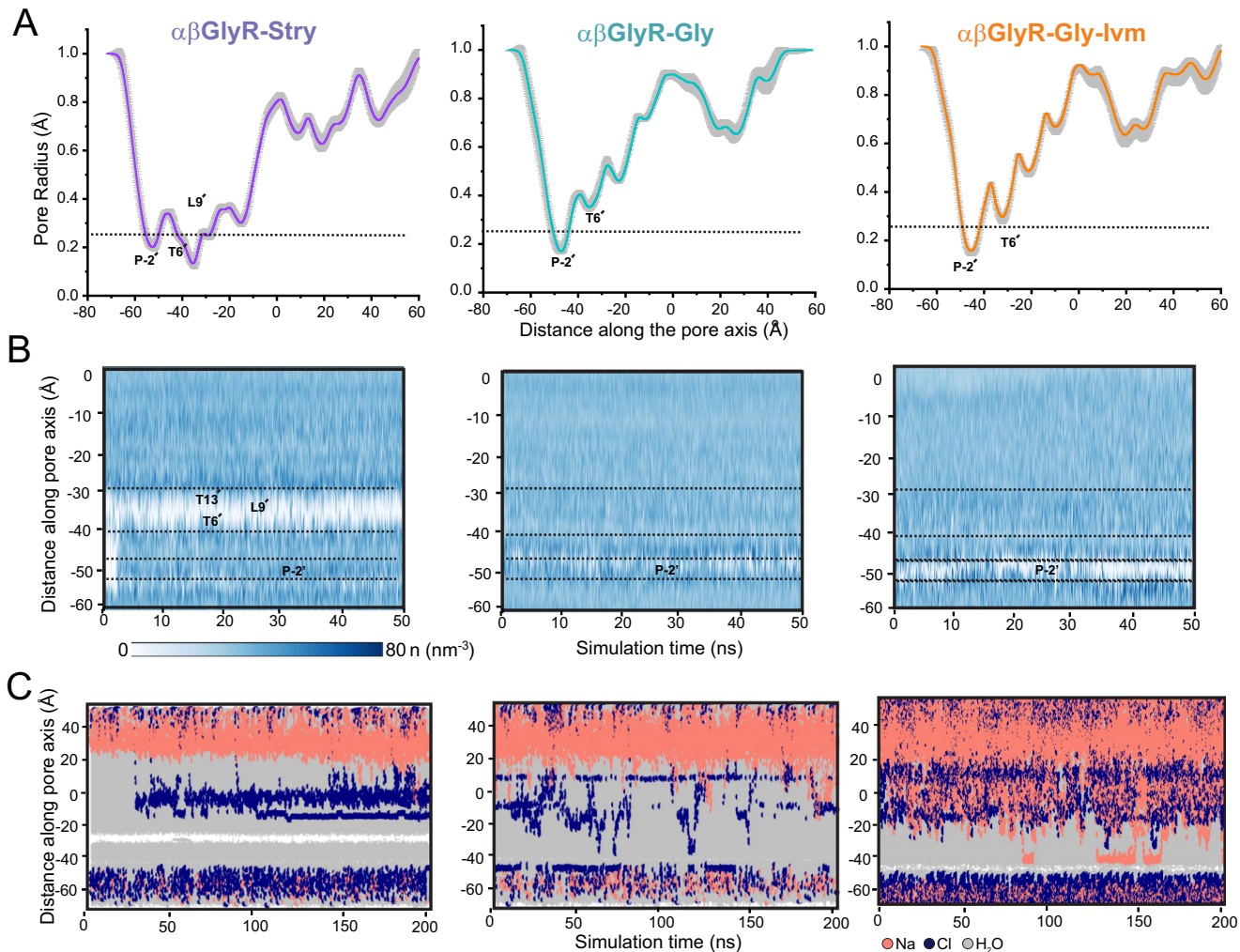

**Fig. 3 | Molecular dynamics simulations of the αβGlyR pore in different conformational states. A** Mean pore radius profile and standard deviations averaged across 50 ns equilibrium simulations for αβGlyR-Stry (left), αβGlyR-Gly (middle), and αβGlyR-Gly-Ivm (right) along the central pore axis. The one-standard-deviation range is shown as a gray band, and the thick line is the mean of a 50 ns simulation. **B** Water density within the channel pore. Each discrete water molecule position is associated with a Gaussian distribution, and the sum of all Gaussians yields the density function of water along the channel center line. Bulk water has a density of 33 nm⁻³. The time series of water density was calculated for 50 ns equilibrium simulations. **C** Trajectories along the pore axis of water molecules and ion coordinates within 10 Å of the channel axis inside the pore, in the presence of a +500 mV transmembrane potential difference (i.e., with the cytoplasmic side having a positive potential). No chloride permeation was observed past L9′ or P-2′ for αβGlyR-Stry, consistent with a closed channel state. No chloride permeation was observed past P-2′ for αβGlyR-Gly and αβGlyR-Gly-Ivm, consistent with a desensitized channel state. During all simulations, positional restraints were placed on the protein backbone, in order to preserve the experimental conformational state while permitting rotameric flexibility in amino acid side chains.

αβGlyR-Gly in αA, Fig. 4C shows the tilt and rotation of the subunit between states, and Fig. 4D shows the difference in main chain atom positions after correcting for tilt and rotation. Overall, rigid rotation of the subunit explains global conformation changes, such as Loop C closure, twisting beta sheets, and TMD rearrangements. A similar analysis of αβGlyR-Gly and αβGlyR-Gly-Ivm structures shows the structures have very small differences in global or individual subunit conformations (Supplementary Tables 6 and 7).

A few areas are not well described by rigid rotation. The complementary interface of the upper ECD shows additional movement toward the neighboring subunit in the αβGlyR-Gly structure. This interface binds a specific class of GlyR PAMs and is also near the epitope for auto-immune inhibitory antibodies, suggesting a mostly unknown role in channel function[33,49]. Another exception to rigid rotation is a small loop that lines the inner channel vestibule (E134-D138 αGlyR, D154-E158 βGlyR). These residues fit snugly within a pocket on the neighboring interface and are essentially unmoved between functional states. They may act as a pivot point during conformational changes.

Local changes also affect the TMD/ECD interface, where the top of the M2 helix is unraveled in αβGlyR-Gly relative to the αβGlyR-Stry. This shifts contact points along the TMD/ECD interface. A distinct feature of this rearrangement in heteromeric αβGlyR is the repositioning of βE318 at the extracellular end of M2. Past research has shown that βE318 is a key contributor to the lower single-channel conductance of heteromeric αβGlyR vs. homomeric αGlyR[31]. In αβGlyR-Stry, βE318 is tucked against the M2–M3 loop of the adjacent primary subunit, interacting with αK300, while in αβGlyR-Gly βE318 is exposed to the pore axis (Supplementary Fig. 6). Similar movements are not seen in βE311, which is also implicated in single-channel conductance differences between heteromeric and homomeric channels, though to a lesser degree than βE318.

## Canonical neurotransmitter binding site
Strychnine and glycine densities are observed at the canonical ligand-binding site. Strychnine density is well-defined in αβGlyR-Stry and is essentially in the same orientation across all five subunits (Fig. 5).

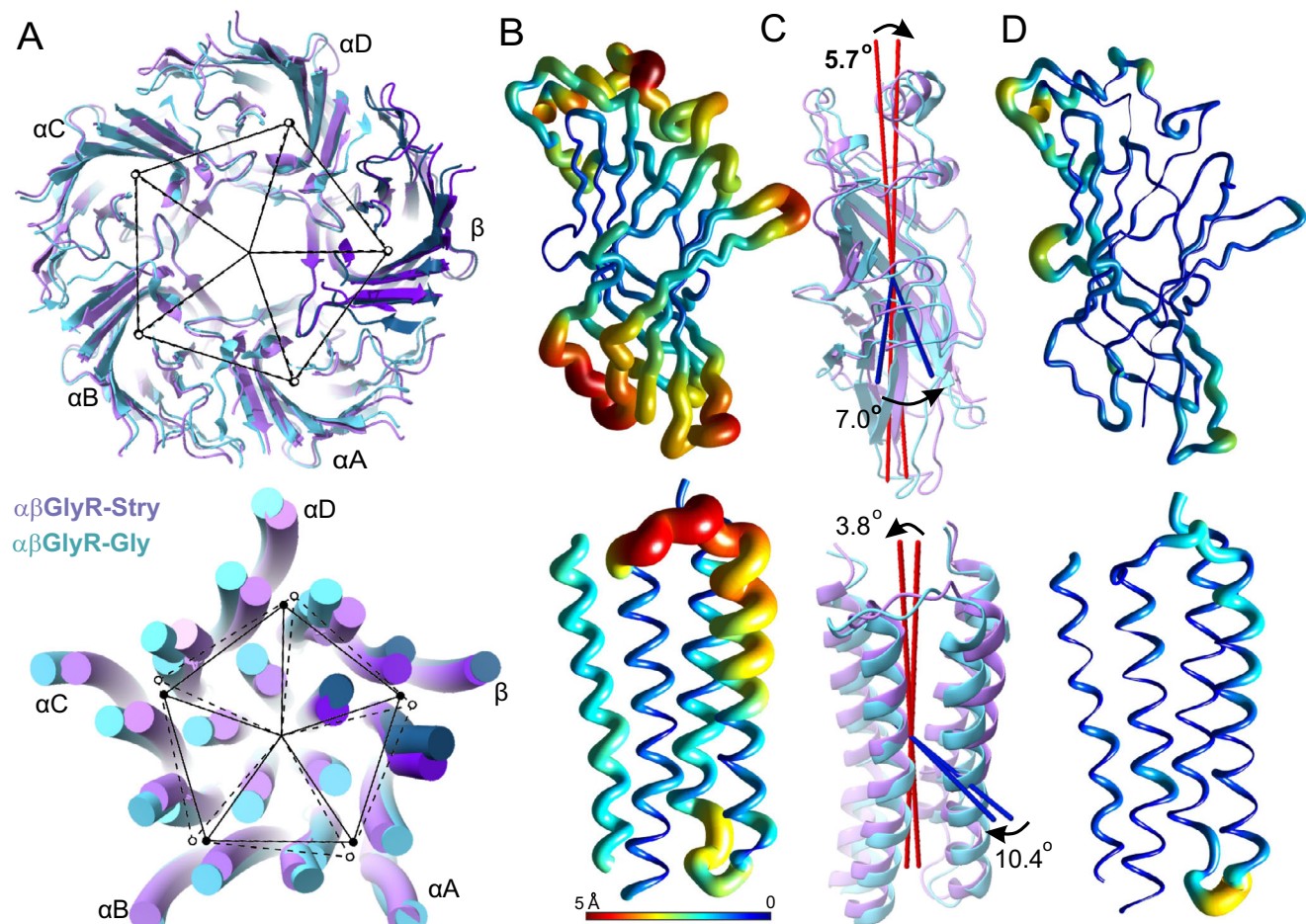

**Fig. 4 | Global conformational differences between αβGlyR-Stry and αβGlyR-Gly.** The ECD and TMD were analyzed separately, with respective results shown in the top and bottom rows. **A** Image overlay showing subunit displacement. The center of mass of each subunit domain is shown as a closed circle for αβGlyR-Stry and an open circle for αβGlyR-Gly. Exact distances and angles between subunits are given in Supplementary Table 6. **B** Ribbon diagrams of the αA subunit. Thickness and color show the displacement of the main chain atom between αβGlyR-Stry and αβGlyR-Gly. Results are similar for other subunits. **C** Images showing the motions described by two of the three principal components. These are associated with subunit tilt (red) and rotation (blue). The measured tilt and rotation are generally consistent between subunits, and exact values for each subunit are given in Supplementary Table 7. **D** Ribbon diagrams showing atomic displacement after correcting for subunit tilt and rotation. Differences highlight conformational changes that are not well-described by the rigid rotation of the subunit domain.

The strychnine rings pack tightly against aromatic residues αF203 (βF203), αF231 (βY252), and α226 (βY246) from the principal subunit and αF87 (βF106) from the complementary subunit. The lactam oxygen is positioned to interact with αR89 (βR108) in the complementary subunit. Glycine density is observed in all five subunits. The carboxyl oxygens form hydrogen bonds with αS153 (βS173) on Loop E, and αR89 (βR108) on Loop D. The glycine nitrogen is positioned to interact with the backbone carbonyl of the phenylalanine on Loop B αF183 (βF203) on Loop B. Strychnine and glycine interactions are summarized in Fig. 5 and are consistent with past homomeric and heteromeric α₁βGlyR structures[12,20,21,23]. However, there is a notable difference in the glycine-binding pocket in h-α₂βGlyR. The phenylalanine on Loop B (αF183 or βF203) is in a distinct orientation, and the glycine nitrogen is facing away from the carbonyl oxygen of the phenylalanine[11]. This may be relevant to the reduced binding affinity of α₂GlyR homomers relative to α₁βGlyR[50].

**Ivermectin binding site**

Ivermectin binds at the ECD/TMD interface between the M3 helix of the principal subunit and the M1 helix of the complementary subunit as characterized in past GluCl (invertebrate glutamate-gated chloride channel) and homomeric αGlyR sturctures[20,21,24–28]. In homomeric αGlyR, this pocket is lined by hydrophobic residues (I249, I253, P254,

and L257 in M1, V304, A312, and L315 in M3) and makes a polar contact with R295 in M2 (from the principal subunit). These interactions are essentially preserved in the α/α interfaces of the αβGlyR-Gly-Ivm. βGlyR introduces differences to the ivermectin binding site at both the α/β and β/α interfaces that are observed in the αβGlyR-Gly-Ivm structure (Fig. 6A and Supplementary Fig. 7A). There is no noticeable difference in the ivermectin pose between the α/α and α/β interface, but there are differences in the polar contacts between ivermectin and surrounding side chains (Fig. 6A). At the α/α interface, hydroxyl oxygen on the benzofuran portion of ivermectin directly interacts with αQ250 on M1 of the complementary subunit, which in turn interacts with αR295 on M2 of the primary subunit. At the α/β interface, βG271 is in the position of αQ250. This allows αR295 on the primary subunit to interact directly with ivermectin. There is also a slight displacement at the top of the primary M2 helix that repositions αGlyR S291 resulting in additional polar contacts not observed at the α/α interface. However, molecular docking scores suggest that the changes in polar contacts within this region are not significant determinants of ligand binding strength. The biggest change between αβGlyR-Gly and αβGlyR-Gly-Ivm is the repositioning of the αA M1−M2 loop, though it is not immediately clear that this is caused by differential ivermectin binding at the α/β interface.

The most notable difference between the β/α interface and α/α interface is βI333 in the position of αA312. The importance of

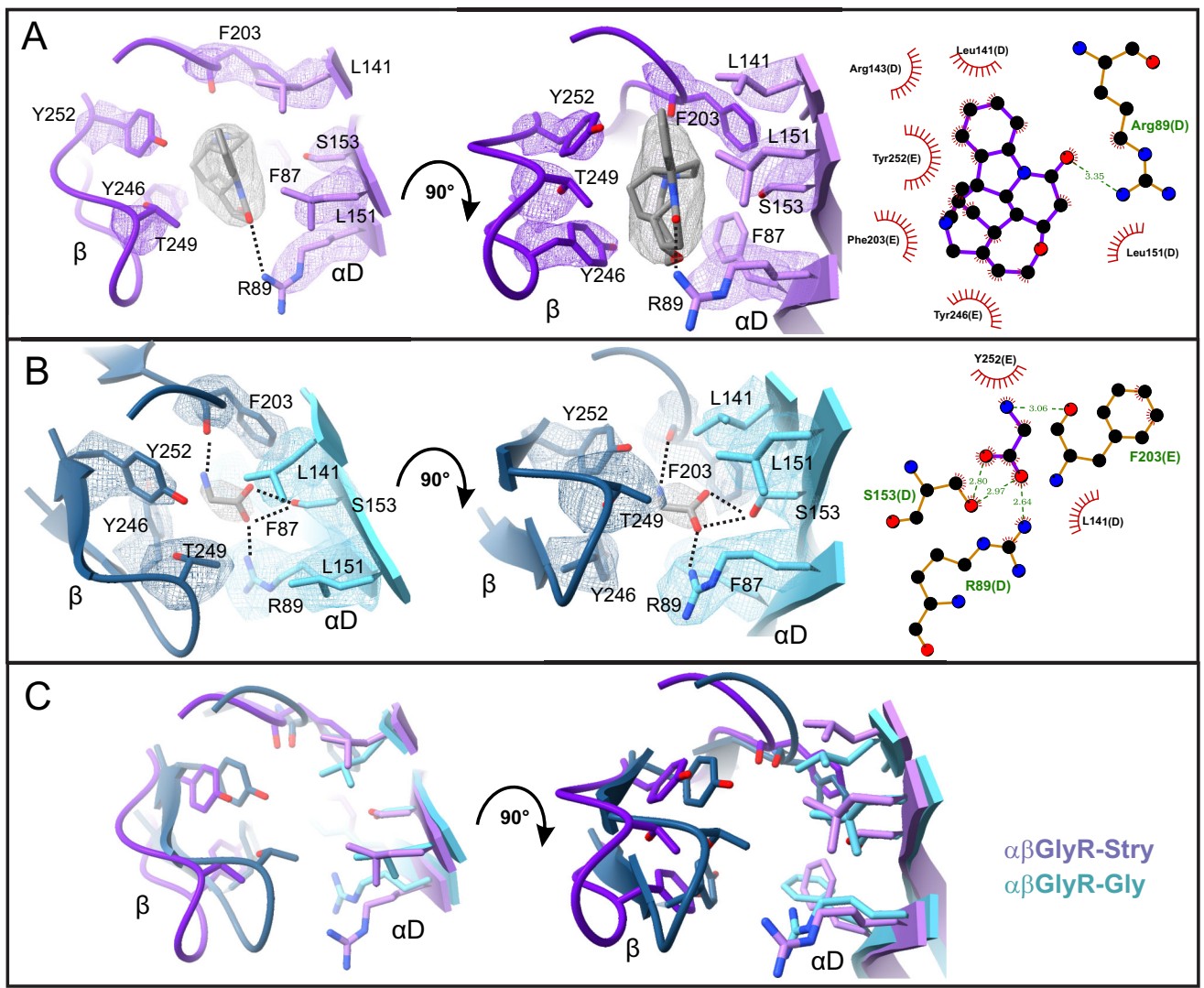

**Fig. 5 | Strychnine and glycine within the canonical binding site.** The binding pocket for **A** strychnine in αβGlyR-Stry and **B** glycine in αβGlyR-Gly shown at different angles. Map density is shown at $\sigma = 0.3$ and $\sigma = 0.55$ for strychnine and glycine, respectively. LigPlot analysis is shown on the right. **C** Overlaid models show structural rearrangements between αβGlyR-Stry and αβGlyR-Gly. Ligands are omitted for clarity.

this position is highlighted in GluCl, which has a glycine at this position. Mutations to alanine shift ivermectin sensitivity in GluCl from nM to μM concentrations, and mutations to phenylalanine render the channel insensitive to ivermectin[26]. However, it was unclear whether this is because bulky side chains prevent ivermectin binding, or whether binding failed to produce an allosteric effect. There is clear ivermectin density at the β/α interface face, but there is a difference in the ivermectin pose compared to other interfaces (Fig. 6A, Supplementary Fig. 7B). The larger side chain of βI333 forces ivermectin away from the primary subunit. This shift is quantified by measuring the distance between the Cα carbon of α A312 (βGlyR I333) and the methyl carbon at position 14 of the ivermectin macrocyclic lactone (Dotted lines Fig. 6A). This distance is 5.0 Å at the α/α or α/β interface and 6.5 Å at the β/α interface. The displacement repositions the rest of the molecule by rotating the methyl carbon at position 12 of the ivermectin macrocyclic lactone by 28° relative to the Cα carbon of αV304 (βV325) (Curved arrows Fig. 6A). The flexible disaccharide is rotated even further away from the pore axis. Docking scores and MD

simulations support that there are differences in the binding energy and pose of ivermectin at the β/α interface (Fig. 6B, C).

## Subunit-specific differences

The overall pentameric assembly is highly symmetric as main chain atoms were within 2 Å of symmetry-related positions in about 90% of modeled residues in each of the three structures. There are, however, several distinct subunit features between αGlyR and βGlyR (Supplementary Movie 1). Density for subunit-specific glycans is observed at each subunit (αN62, βN54, and βN241), which allows for unambiguous subunit assignment (Supplementary Fig. 8A, B). All three cryo-EM maps show a βGlyR-specific density near the N-terminal (Supplementary Fig. 8B, C). The density is clearest in αβGlyR-Stry and αβGlyR-Gly-Ivm where it runs anti-parallel to the N-terminal α-helix before descending into the channel vestibule. This density was modeled as a βGlyR-specific N-terminal extension (βP40-βS53), consistent with the observation that βGlyR has an additional 21 N-terminal residues relative to αGlyR. αGlyR subunits have a post-M4 density that extends up to the β9 strand of the ECD. This density corresponds to 10 additional

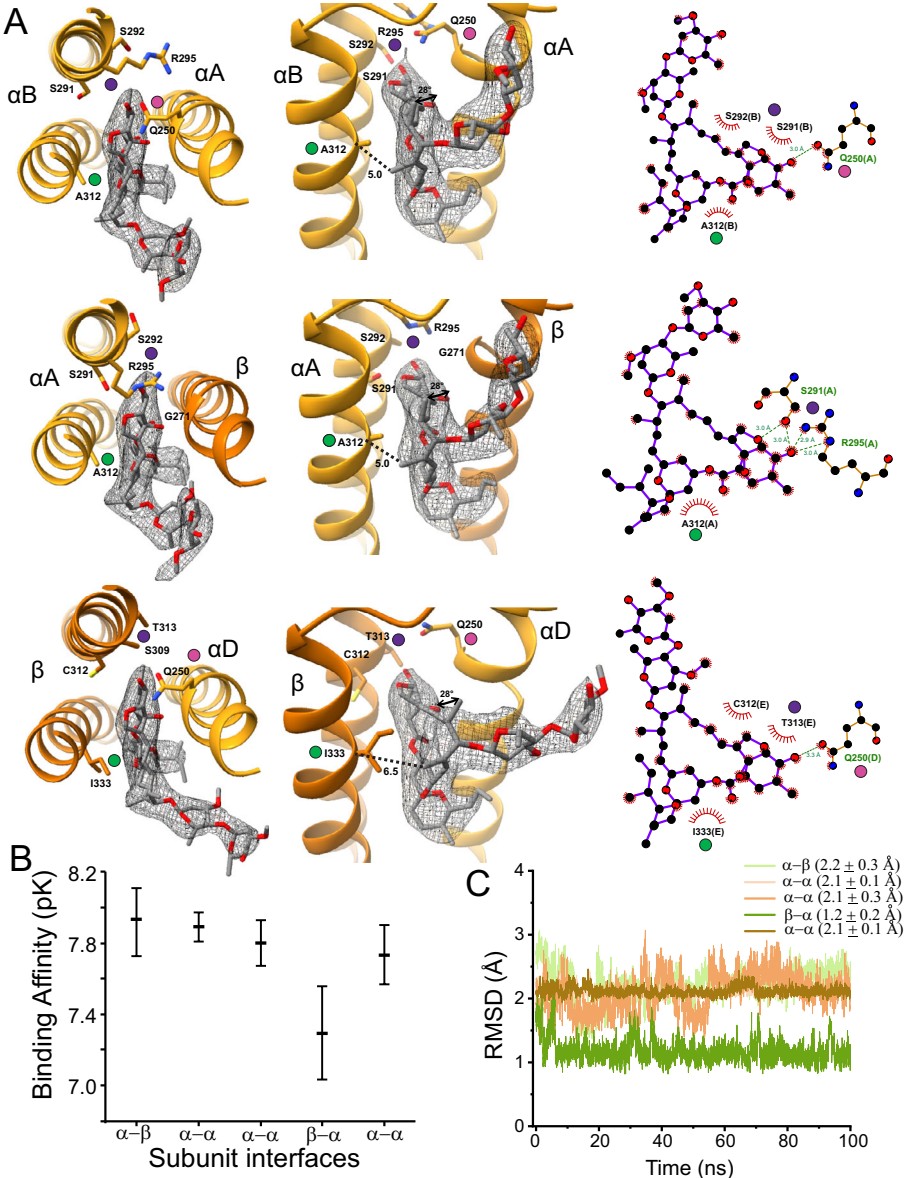

**Fig. 6 | Subunit-specific properties of ivermectin binding. A** A select set of ivermectin interactions is shown at different angles and with LigPlot results. Map density is shown at $\sigma = 0.11$. The αB/αA, αA/β, and β/αD interface are respectively shown in the top, middle and bottom rows. Colored dots show the same location in different views. The dotted lines show the distance from the Cα carbon at αA312 (βI333) to the methyl carbon at position 14 of the ivermectin macrocyclic lactone. The curved arrows in the middle panel of α/β show the rotation, relative to αV304 (βV325), of the methyl carbon at position 12 of the ivermectin macrocyclic lactone.

For clarity, only the side chains discussed in the manuscript are shown, and the full interaction profile is given in Supplementary Fig. 7. **B** Binding affinity of ivermectin binding to different subunit interfaces evaluated with the GNINA docking scoring function. The mean and standard deviation are from an ensemble of models as described in ref. [99]. **C** Root mean square deviation (RMSD) of ivermectin at different subunit interfaces during an unrestrained simulation. The RMSD is calculated with respect to the cryo-EM binding pose.

post-M4 residues in αGlyR (αI435 to αQ444), which were best observed and modeled in αβGlyR-Gly-Ivm (Supplementary Fig. 8D). The cryo-EM map for both αβGlyR-Stry and αβGlyR-Gly-Ivm has a number of TM-associated lipid densities, as previously observed in other GlyR and pLGIC structures[12,21,22,51,52]. There are subtle differences between the α/α, α/β, and β/α interfaces (Supplementary Fig. 8D). A similar pattern is observed in the lipid densities of the native α₁βGlyR structure[12]. These differences are perhaps relevant to the differential effect of neurosteroids on homomeric and heteromeric GlyR[53].

### Structure of the intracellular domain
The ICD of GlyR is comprised primarily of the cytoplasmic loop that connects the M3 and M4 helices and is made of 80 residues in αGlyR

and 118 residues in βGlyR. It is mostly unstructured, but through partial signal subtraction and focused local refinement of αβGlyR-Stry particles, extensions of the M3 helix were partially resolved for each of the subunits (Fig. 7). AlphaFold[54] predicts that this region (αH335-αG354 and βP356-βA372) is α-helical for both αGlyR and βGlyR, but the helix in βGlyR is tilted relative to the M3 helix at position βP356 (αH335). Consistent with this prediction, the cryo-EM density extends downward near αGlyR subunits, but the density near the βGlyR subunit is bent about 45° towards the neighboring αD subunit. The M3 extensions of each subunit seem to intertwine and eventually converge to a merged density that is tilted about 30° away from the C5 axis, towards αB. The density of the four αGlyR post-M3 regions is distinct, suggesting there is conformational heterogeneity within the ICD, even

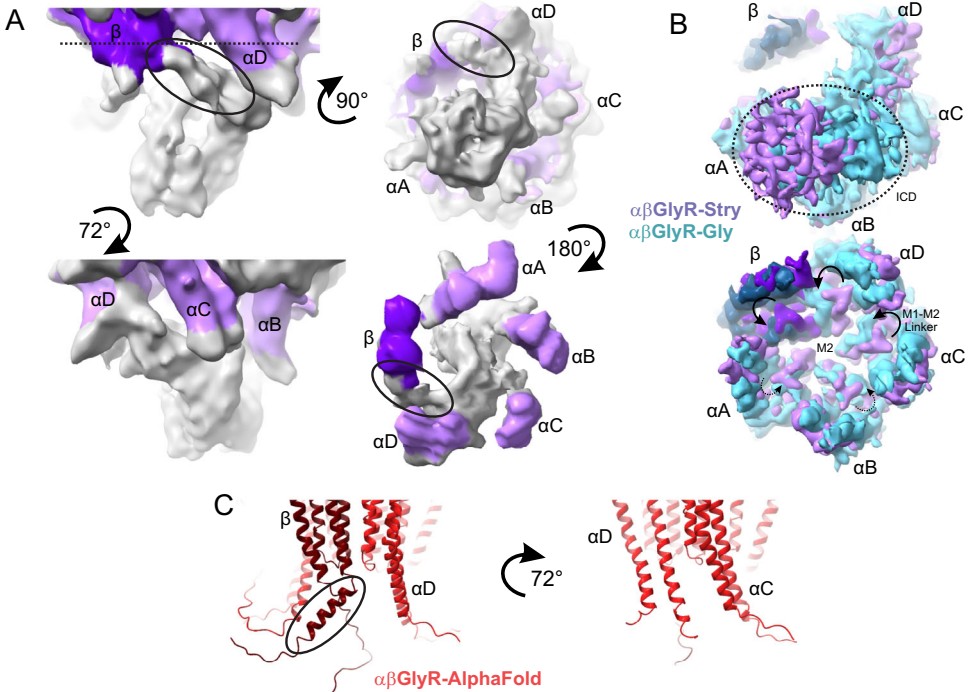

**Fig. 7 | Structural characterization of αβGlyR ICD. A** Partial signal subtraction and local refinement of αβGlyR-Stry particles reveal general features of the ICD. The density is shown at σ = 0.09 at multiple angles. Map density is colored purple near residues modeled in the full αβGlyR-Stry map, with the darker shade corresponding to βGlyR. Density is colored gray for regions not modeled. The black oval shows an asymmetric bend in the density βGlyR. **B** Bottom view of the full map of αβGlyR-Stry and αβGlyR-Gly. The top image is shown at a low threshold and the bottom at a high threshold (σ = 0.133 and σ = 0.17 for αβGlyR-Stry, σ = 0.23 and σ = 0.33 for and αβGlyR-Gly). The dotted circle shows the ICD density in the low threshold image, and the arrows show M1−M2 loop displacement in the high threshold image. M1−M2 displacement seems less pronounced in subunits near the ICD density. **C** AlphaFold predictions of the TMD and ICD of the assembled heteromeric channel. The left image shows the interface that matches β/αD and the right shows the interface that matches αD/αC. The black oval shows a break in the post-M3 helix of βGlyR that occurs near the asymmetric density highlighted in panel (**A**). However, the predicted bend is in the opposite direction of the observed density.

among compositionally identical α subunits. Similar treatment of αβGlyR-Gly or αβGlyR-Gly-Ivm particles shows features that were consistent with the αβGlyR-Stry ICD but more poorly resolved. Comparison of the position of the ICD density in the full map of αβGlyR-Stry and αβGlyR-Gly reveals an asymmetric positioning of this region with respect to the pore axis, placing it beneath αA and αB subunits. Perhaps as a consequence of this ICD orientation, the M1−M2 linker of αA and αB subunits are less displaced during glycine-induced activation compared to αB, αC, and β subunits. The ICD may influence asymmetric gating movements within the M2 helices via the M1−M2 linker.

## Discussion

The presented work has important similarities and distinctions with other recently reported heteromeric GlyR structures. A key similarity is further evidence that the 4α:1β stoichiometry is predominant, if not exclusive heteromeric GlyR stoichiometry. There are small conformational differences between the structures in this work and related heteromeric GlyR structures (Supplementary Fig. 9). Comparing αβGlyR-Stry to the two strychnine bound states of $\alpha_2\beta$ heteromeric GlyR in Yu et al.[11] (PDB 7L31, 7KUY) we observe that αβGlyR-Stry is more constricted at the −2′ position. This may be due to differences in h-$\alpha_2\beta$GlyR and ZF-$\alpha_1\beta$GlyR, the expression system, or ICD truncation in the $\alpha_2\beta$GlyR sample. There is general agreement between αβGlyR-Gly and other published glycine-bound desensitized states ($\alpha_2\beta$ heteromeric GlyR[11], PDB 5BKF, and native $\alpha_1\beta$ heteromeric GlyR[12], PDB 7MLY), though we note a few key differences. In αβGlyR-Gly, the desensitization gate is slightly more asymmetric because the αA and αD P-2′ residues are positioned radially outward compared to corresponding residues in other models. The cytosolic end of the M4 helix is also positioned radially outward in αβGlyR-Gly compared to the other

models. This region binds lipid and neurosteroid modulators, but given the limitations of detergent and saposin protein extraction, one cannot confidently assign physiological significance to differences in lipid-binding domains. It is also noteworthy that a glycine-bound open conformation was also obtained from $\alpha_2\beta$ heteromeric GlyR data in Yu et al.[11] (PDB 5BKG), whereas both αβGlyR-Gly and native $\alpha_1\beta$ heteromeric GlyR were only found in desensitized conformations. The pore is also different at the 6′ position between our work and others, consistent with species-specific leucine/phenylalanine at this position. Finally, Zhu et al.[12] found that the $\alpha_1(+)/\beta(-)$ interface has the most buried surface area in the glycine-bound state, whereas this interface was the most loosely packed in our structure. This may be related to species-specific differences, components extracted from native tissue, or the antibodies used as fiduciary markers.

A distinct feature of this work is the ivermectin-bound structure and in particular, visualizing the ivermectin-binding site at different subunit interfaces. The ivermectin pose at the α/α and α/β interface is nearly identical to those observed in past homomeric GlyR structures[21,24]. There are differences in the observed hydrogen bonding network between α/α and α/β, though both computational docking and past electrophysiology studies suggest that these are not large determinants of ivermectin activity[27]. Ivermectin binding at the β/α interface is clearly distinct due to the effects of βI333, consistent with past functional mutagenesis[26]. This is further emphasized by comparing the ivermectin density of each of the heteromeric GlyR interfaces to that of GluCl and homomeric α1GlyR[21,25] (Supplementary Fig. 7C). Compared to the α/α or α/β interface, the ivermectin in GluCl is shifted even closer to the M2/M3 helices in the principal subunit, as expected given GluCl has a glycine at the A312 position. The distance between G281(GluCl)/A312(αGlyR) and the methyl carbon at position

14 of the ivermectin macrocyclic lactone is 4.3 Å in the GluCl ivermectin model (PDB 3RIA) and 5.7 Å in full-length homomeric GlyR (PDB 6VM3). For αβGlyR-Gly-Ivm, these distances are 5.0 Å and 6.5 Å at the α/α or β/α interface, respectively. Further functional and structural studies may help elucidate what elements are critical for ivermectin potentiation and activation of GlyR and how these actions occur.

Our work also advances the field by characterizing additional portions of βGlyR at the N-terminal ECD and ICD. The N-terminal density of βGlyR is also resolved in other GlyR maps, though it was attributed to the β-specific glycosylation of N54. In our opinion, the density is more consistent with that of a peptide chain, an interpretation that is corroborated by AlphaFold predictions[54]. Though conserved across species, the role of this β-specific sequence has not, to our knowledge, been studied in detail. However, the residues that our model puts adjacent to this region have been implicated in channel assembly[55]. Beyond what we have modeled, there are 12 additional N-terminal residues, of which 8 are lysine. This large concentration of positive charge, likely in the channel vestibule, has interesting implications for channel assembly and function.

Though mostly structurally uncharacterized, the ICD plays a critical role in channel function and cellular localization. The low-resolution features of the αβGlyR ICD-focused map can be compared to two well-characterized ICD features of cationic pLGICs[56,57]. These are the post-M3 MX helix, which is amphipathic and mostly parallel to the cytosolic membrane interface, and the pre-M4 MA helix, which descends from the membrane with each subunit joining a helix bundle along the pore axis. The αβGlyR ICD density suggests that the post-M3 region of each GlyR subunit comes together below the channel. Though comparable density is not observed in homomeric GlyR structures, AlphaFold and circular dichroism studies suggest the post-M3 region of αGlyR is α-helical[54,58]. This feature may be a helical bundle analogous to the MA helical bundle in cationic channels. A defining feature of the 5-HT$_{3A}$R MA helix is positively charged residues that line lateral portals and limit cation conductance[59]. Positively charged residues that enhance chloride conductance are found on the analogous pre-M4 portion of αGlyR[60]. However, a large number of positive charges are also concentrated in the post-M3 portion of αGlyR and βGlyR (αR340-αK349, βR363-βK371). It is likely that positive charges in both the post-M3 and pre-M4 portions of αGlyR contribute to channel conductance. Unlike cationic pLGICs, the ICD αβGlyR density is off-axis, which we hypothesize is due to the asymmetry of the post-M3 portion of βGlyR. This portion of βGlyR is predicted by AlphaFold to be a bent helix, similar to the MX helix. The αβGlyR ICD density indeed bends near the β subunit, though in a different direction than Alpha-Fold predicts. The MX helix appears in different conformations across functional states, and the energetics of its insertion in and out of the membrane is thought to modulate channel activity[56]. It is possible a βGlyR bent helix plays a similar role. It may also impose some structural restraints not present in the ICD of homomeric αGlyR.

The immediate post-M3 region of αGlyR includes an arginine-rich portion (αR340-αK349). Protein interactions within this region are critical to protein folding and trafficking[61,62]. Additionally, Gβγ dimers directly interact with this sequence, mediating the effects of ethanol potentiation of GlyR at behaviorally relevant concentrations[63–65]. Notably, a helical structure is necessary for Gβγ-mediated ethanol effects[58]. Just prior to the arginine-rich region, αL338 and αL339 have been implicated in PKC-mediated endocytosis of homomeric α$_1$GlyR[66]. Interestingly, this effect is not observed with heteromeric α$_1$βGlyR, supporting the idea that the presence of βGlyR affects the ICD structure of αGlyR subunits. An α$_3$GlyR-specific PKA phosphorylation site is located near the end of the AlphaFold predicted post-M3 helix[67]. This site is associated with PGE$_2$-induced pain sensitization, and its phosphorylation may collapse the ICD structure, leading to channel inhibition. Absent from the observed ICD density are the gephyrin-binding domain[8] and a pre-M4 poly-proline helix. The poly-proline helix

contains an SH3-binding domain that interacts with syndapin I and possibly other proteins[68,69].

The cumulative results of this project outline structural features of heteromeric GlyR that result in distinct conformational properties relative to homomeric αGlyR. Additionally, the αβGlyR-Gly-Ivm structure demonstrates asymmetric ivermectin binding based on subunit heterogeneity. The expression and purification strategy also provides a protocol for further structural work that may be more amenable than native tissue extraction[12]. Ultimately, we expect this work will contribute to the evolving paradigm of this biologically and clinically relevant channel.

## Methods

### α$_1$β-GlyR expression in HEK 293 T cells and electrophysiology studies

ZF-α$_1$GlyR and ZF-β$_B$GlyR were cloned into pcDNA3.1(+)-P2A-eGFP and pICherryNeo as described in Supplementary Tables 1 and 2. HEK 293 T cells were grown in a 10 cm dish to ~80% confluency, at which point they were transfected with 2.5 μg of ZF-α$_1$GlyR DNA and 12.5 μg of ZF-β$_B$GlyR DNA using 45 μg of Mirus TransIT-2020 transfection reagent (Mirus MIR 5404). Twenty-four hours later, cells were detached from the 10 cm plate using 5 mL of Accutase (Innovative Cell Technologies AT 104) for 5 minutes, spun down, and then resuspended in 293 SFM-II media (Gibco 11686029) supplemented with 20 mM HEPES and 1% Pen/Strep for 1 h.

An IonFlux Mercury 16 instrument (Fluxion Biosciences) and IonFlux 16 software (v5.0) were used for electrophysiology experiments. Extracellular solution (ECS) was composed of 138 mM NaCl, 4 mM KCl, 1 mM MgCl2, 1.8 mM CaCl2, 5.6 mM glucose, and 10 mM HEPES at pH 7.4 and 298 mOsm. Intracellular solution (ICS) was composed of 70 mM KF, 60 mM KCl, 15 mM NaCl, 5 mM EGTA, and 5 mM HEPES at pH 7.2 and 283 mOsm. Glycine 100 mM stock solution was prepared in ECS and diluted to yield 10, 25, 30, 50, 75, 100, 300 μM, and 1 mM solutions. The resuspended HEK 293 T cells and solutions were added to the appropriate wells of a 96-well ensemble IonFlux Plate. Each plate can accommodate 12 separate experiments and perfuse up to 8 different solutions per experiment. Cells were attached to electrodes within the 'trap' well per the manufacturer's guidelines. Ensemble currents were then recorded from each trap at 1000 kHz. The recording was done over multiple sweeps that covered the prepared glycine concentrations. During each sweep, cells were held at a −60 mV potential for 400 ms. A 200 ms pulse from −60 to −70 mV was then performed to assess patch resistance. 1400 ms after return to baseline holding potential, solutions from the compound wells were administered for 3 s. Solutions were then washed out for 10 s over the remainder of the sweep. Data were plotted and analyzed using ClampFit (version 11.2; Molecular Devices), Origin (version 9.9.0.225; OriginLab Corporation), and Excel (Microsoft, v16.6) software.

### α$_1$β-GlyR expression and purification for structural studies

Zebrafish full-length GlyR proteins and rat gephyrin domain E (Geph-E residues 398–769) were used in the present study (NCBI reference sequence: GlyRα$_1$ NP_571477.1, GlyRβB NP_001003587.1, Geph-E NP_074056.2). ZF-α$_1$GlyR, ZF-β$_B$GlyR, and rat Geph-E have 86%, 85%, and 100% sequence homology to corresponding human genes, with the ICD region of GlyR showing the most species-specific divergence. GlyR constructs were designed with a C-terminal protease sequence (Thrombin, LVPRGS, for α$_1$GlyR and TEV, ENLYFQG, for β$_B$GlyR) followed by an affinity tag (8x His tag for α$_1$GlyR and 1D4 tag, TETSQVAPA, for β$_B$GlyR). β$_B$GlyR also included multiple N-terminal Strep tags, which were not used for purification. An N-terminal flag tag (DYKDDDDK) was placed on Geph-E followed by a poly-asparagine linker and TEV cleavage site. GlyR genes were codon optimized and cloned into pFastBacDual vector with the α$_1$GlyR subunit under the p10 promoter

and the GlyRβ subunit under the polyhedrin promoter. This approach prevents us from changing the relative expression of each subunit but does increase consistency from prep to prep. The Geph-E gene was similarly codon-optimized and cloned into the pFastBac1 vector.

Baculovirus was generated using the Spodoptera frugiperda (Sf9) cells[70] expression system using a pFastBacDual vector that carried both GlyR genes or pFastBac1 for Geph-E. In either case, the measured titer was about $3 \times 10^8$. Two liters of ExpiSf9 cell culture (purchased from Invitrogen) were seeded at a density of 2–2.5 mil/mL and infected the next day with 2 and 1.6 mL/L of GlyR and Geph-E virus, respectively. Protein purification was carried out as previously described[70]. Briefly, cells were harvested 48 h post-infection, pelleted, and resuspended in lysis buffer, (20 mM tris-base, 36.5 mM sucrose, 10% glycerol, and 0.25% Sigma 8340 protease cocktail inhibitor, pH 8.0), and flash frozen. Thawed cells were then lysed by gentle sonication on ice and centrifuged at $3200 \times g$ for 15 min to remove large cell debris. The supernatant was then centrifuged at $167,000 \times g$ for 1 h, and the membrane pellet was resuspended in membrane buffer (20 mM HEPES, 150 mM NaCl, 10% glycerol, pH 8.0). The resuspended membrane was flash-frozen and stored until use. The thawed membrane resuspensions were solubilized for 2 h at 4 °C in 15 mM n-dodecyl-β-D-maltopyranoside (DDM, Anatrace) and supplemented with 0.05% CHS (Anatrace) and 0.05 mg/mL soybean polar extract (Avanti Polar Lipids). The mixture was then centrifuged at $167,000 \times g$ for 20 minutes, and the supernatant was taken to remove the non-solubilized fraction. The solubilized fraction was subjected to two rounds of affinity purification. The first step used CNBr-activated sepharose beads (Cytiva 17-0430-01) conjugated to 1D4 antibody, pre-equilibrated in wash buffer (20 mM HEPES, 150 mM NaCl, 1 mM DDM, 0.05% CHS, pH 8.0). The beads were mixed with the solubilized protein, rocked at 4 °C for 2 h, and then washed with 10 column volumes of wash buffer. The beads were then eluted in 2 column volumes of 1D4 elution buffer (wash buffer + 4 mg/mL of 1D4 peptide, generated by Genscript: TETSQVAPA, pH 8.0). After 2–3 h, the first elution volume was collected and replaced with 2 additional column volumes of 1D4 elution buffer. The next day the overnight elution volume was collected, and both elution volumes were combined and bound to pre-equilibrated Nickel-NTA agarose beads (Qiagen 30210) at 4 °C for 2 hours. The beads were washed with 5-column volumes of wash buffer containing 25 mM imidazole at pH 8.0 and then eluted for ten minutes in the presence of a 1-bed volume of an elution buffer containing 250 mM imidazole at pH 8.0. The elution step was repeated 5× total. The elution volumes were then recombined and concentrated in a 100 kDa MWCO Millipore filter (Amicon UCF810024), and the protein concentration was measured using nanodrop. The solution was filtered with a 0.22 micron PVDF filter and then passed through a Superose 6 Increase column (Cytiva 29091596), pre-equilibrated with filtration buffer (20 mM HEPES, 150 mM NaCl, 1 mM DDM, pH 8.0). Western blot analysis showed the peak fraction contained both GlyR subunits and Geph-E. This fraction was used for cryo-EM imaging. Purification of αβGlyR without Geph-E produced a peak that eluted -0.8 mM to the right. The fraction concentration was between 0.05–0.1 mg/mL, and the sample was not concentrated following gel filtration.

## Mass photometry

Mass and relative abundance of samples were measured using a TwoMP mass photometer (Refeyn Inc., Oxford, UK) as described before[71]. Briefly, for sample loading, silicone gaskets (CultureWell™ reusable gasket, 3 mm diameter × 1 mm depth, Grace Bio-Labs) were placed onto clean glass coverslips (Refeyn Inc.). Data were acquired for 1 min using AcquireMP v2022 R1 software. In each well, 18 μl of buffer was added, and data were collected first. Subsequently, a 2 μl protein sample was mixed with the buffer in the well to a final concentration of -10 nM. Data were analyzed using DiscoverMP v2022 R1 software to assign mass and generate histograms. The molecular weight of the samples was assigned by contrast comparison of known mass standard calibrants measured on the same day.

The sample yielded peaks at 44, 86, 396, 804, and a very small peak at 1121 kDa. The 44 kDa peak is slightly smaller than a detergent micelle but was similar to what was observed in a sample containing only 1 mM DDM. The predicted molecular weights for each gene, including affinity tags, are 49.8, 63.2, and 49.6 kDa for αGlyR, βGlyR, and Geph-E, respectively. The predicted molecular weight for a 4α:1β complex bound to a Geph-E dimer is 361.6 kDa. Adding 44 kDa for the detergent micelle gives 405.5 kDa. Given the uncertainty of the detergent mass surrounding GlyR, this is reasonably close to the expected measured mass of 396 kDa. A homomeric assembly of αGlyR would be expected to have a molecular weight of 293.1 kDa.

## Cryo-EM sample preparation and imaging

Graphene-oxide (GO) coated grids were prepared from Cu 300 R1.2/1.3 holey carbon grids (Quantifoil, Micro Tools) using the drop-cast method[72]. A 0.2 mg/mL GO solution was prepared from a stock of 2 mg/mL GO solution diluted in MilliQ water (Sigma Aldrich, 763705), followed by centrifugation at $300 \times g$ for 1 min to remove large GO aggregates. EM Grids were glow discharged at 30 mA for 30 s, and then 3.5 μL of the GO solution was applied to the carbon-coated side for 1 min. The solution was gently blotted away with filter paper, immediately placed on a water droplet, and then blotted again to remove the water. The water droplet wash was done one additional time on the carbon side and once on the non-carbon side. The resulting grids had about 50% of the holes covered with a thin layer of GO.

Prior to sample application, grids were glow-discharged in the range of 5 mA for 5 s to 20 mA for 20 s. The following ligands were added to the sample about 1 h prior to freezing: 1 mM glycine for αβGlyR-Gly, 1 mM, and 20 μM ivermectin (Sigma Aldrich, stock solution prepared by dissolving in a final concentration of 0.1% DMSO) for αβGlyR-Gly-Ivm, and 100 μM strychnine (Sigma Aldrich) for αβGlyR-Stry. For αβGlyR-Gly-Ivm, ivermectin was added 10 min prior to glycine, and 1 mM fluorinated Fos-Choline-8 (Anatrace) was added to the sample. The protein sample was applied and blotted twice prior to plunge freezing in liquid ethane using a Vitrobot Mark IV.

Imaging was done using a 300 kV FEI Titan Krios microscope equipped with a K3 camera and a Gatan Imaging Filter (GIF) at either New York Structural Biology Center (NYSBC, αβGlyR-Stry) using Leginon (v3.5) or at Case Western Reserve University using (CWRU, αβGlyR-Gly, and αβGlyR-Gly-Ivm structures) using EPU (v2.7). For αβGlyR-Gly, data were collected in super-resolution mode at 81 K magnification. The physical pixel size was 1.1 Å/pixel with a total dose of 60 e⁻/Å². 6059 movies were collected in correlated double sampling (CDS) mode with 50 frames/movie, and 5808 movies were collected without CDS mode with 40 frames/movie. For αβGlyR-Stry, 14,270 movies were collected in counting mode at 105 K magnification. The physical pixel size was 0.825 Å/pixel with a total dose of 60 e⁻/Å² collected in 50 frames without the CDS mode. For the αβGlyR-Gly-Ivm dataset, 10,036 movies were collected in counting mode at 105 K magnification. The physical pixel size was 0.84 Å/pixel with a total dose of 60 e⁻/Å² collected in 50 frames without the CDS mode. In each case, defocus values were set from −0.8 to −1.8 μm.

## Image processing

Motion correction was done using RELION implementation of the MotionCor2 (v1.2.3)[73] algorithm with a B-factor of 150 pixels[2] (Initially RELION v3.1[74], later v4.0). For αβGlyR-Gly, super-resolution images were binned (2 × 2) in Fourier space. CTF estimation was done using CTFFIND (v4.1)[75] and GCTF (v1.06)[76]. Particle picking was done in RELION, initially using a template from α1GlyR[21] and later repicked using a template generated from an early 3D refinement. The final round of picking generated 3–4 million particles for each dataset which then went through -10 rounds of 2D classification in RELION,

transitioning from particles binned 4×, then 2×, and then finally at the original pixel size. A final round of 2D classification was done in cryoSPARC v3.3.1[77]. Transitions between cryoSPARC and RELION were done using pyem software (v0.5)[78]. A subset of 190296, 357696, and 298,446 particles was used for initial reconstructions of αβGlyR-Stry, αβGlyR-Gly, αβGlyR-Gly-Ivm maps, respectively.

Using cryoSPARC, an ab initio reconstruction was done with the particles from 2D classification. The reconstruction was then used for a non-uniform (NU) refinement with imposed C5 symmetry, which was then used for a NU refinement with no symmetry constraints. This refinement was then used for Bayesian polishing in RELION, followed by 3D classification using symmetry relaxation[79]. This led to two classes, one clearly identifiable as a 4α:1β structure based on subunit-specific glycosylation, while the other showed a C5 structure with mixed α/β features. Using the class with distinct subunit features as a reference, all particles were refined to a single population with the 4α:1β stoichiometry. Particle sets were then curated with multiple rounds of NU refinement, Bayesian polishing, and 2D/3D classification without image alignment or with local angular refinement and symmetry relaxation. Symmetry relaxation allows particles to sample all five pseudo-symmetric orientations. Though this may scramble particles in the wrong orientation, our expectation is that more and more particles would find the right orientation as the structure progressed. High-resolution classes were selected, and maps with the C1 features shown in Supplementary Fig. 8 were used as templates for further refinement. Care was taken to sufficiently low-pass input templates to avoid noise bias. No other subunit combinations were observed throughout the process, though without fiduciary markers, one cannot make a definitive statement about other possible combinations in low-resolution classes. The final refinements included 84,437, 99,183, and 204,512 particles at 3.0, 3.2, and 3.0 Å resolution for αβGlyR-Stry, αβGlyR-Gly, αβGlyR-Gly-Ivm, respectively. None of them resulted in clearly defined alternate conformations. Local resolutions (2–8 Å) were estimated using the RESMAP software (1.1.4)[80].

The final particles from the αβGlyR-Stry dataset were further explored by partial signal subtraction followed by local refinement. Multiple rounds were done with masks that progressively removed more of the TMD (Supplementary Fig. 2). The mask that gave the best ICD visibility covered about 2/3 of the TMD. Similar treatment of αβGlyR-Gly or αβGlyR-Gly-Ivm particles gave ICD maps with similar but less well-defined features.

## Model building

Unsharpened cryo-EM maps were used for model-building within density for the entire ECD, TMD, and a small region of the ICD. A homology model of αβGlyR was built using AlphaFold[54] for individual subunits aligned with a previous glycine-bound homomeric α1GlyR structure[21]. The homology model was then fit to the αβGlyR maps using stepped refine and manual editing in Coot (v0.9.4.1)[81]. The structures were further refined using the phenix.real_space_refinement tool from the PHENIX software package (v1.19.1-4158)[82,83], using rigid body, local grid, and gradient minimization. The individual models were then subjected to additional rounds of manual model fitting and refinement. The refinement statistics and the final model to map cross-correlation were evaluated using the PHENIX module mtriage, and the stereochemical properties of the models were evaluated by Molprobity[84] (results are given in Supplementary Table 5). The pore profile was calculated using the HOLE program (v3.0)[85]. Figures were prepared using ChimeraX (UCSF, v1.11, 1.11.2)[86] and PyMOL (Schrödinger, LLC, v.2.0.4) and CorelDraw (v.20.1.0.708). PCA was done in MATLAB (MathWorks, v.R2018b). Ligand analysis was done using Lig-Plot (v 4.5.3) PCA was done by pairing two conformations and defining 6 degrees of freedom for each atom, the xyz positions in one conformation, and the displacement between the two conformations. In each case, 3 of the 6 PCA vectors described position-dependent

displacements, i.e. rigid rotations, and the remaining 3 were mostly position independent, i.e., local deformations.

## MD simulations

Cryo-EM structures of the αβGlyR in each state were embedded within phospholipid (POPE) bilayer membranes with the CHARMM-GUI Membrane Builder[87,88] in $10 \times 10 \times 17$ nm³ simulation cells. Simulations were performed with GROMACS (2021)[89], using the TIP3P water model[90] and the CHARMM36m forcefield[91] with pair-specific Lennard-Jones parameters to correct the chloride interactions with proteins, lipids, and alkali cations[92]. The integration time-step was 2 fs. Bonds were constrained through the LINCS algorithm[93] implemented in GROMACs (2021). A verlet cut-off scheme was applied, and long-range electrostatic interactions were calculated using the Particle Mesh Ewald method[94]. Temperature and pressure were maintained at 310 K and 1 bar during simulations, using the velocity-rescaling thermostat[95] in combination with a semi-isotropic Parrinello and Rahman barostat[96], with coupling constants of 1 and 5 ps, respectively.

Pore water-free energy profiles were computed for alternative conformations of the protein using the Channel Annotation Package[44], in each case based on three replicates of 50 ns equilibrium simulations at physiological salt (150 mM NaCl) concentration. To preserve the conformational state of each cryo-EM structure during simulations, harmonic restraints at a force constant of 1000 kJ/mol/nm² were placed on protein backbone atoms. Simulation trajectories were analyzed at 100 ps intervals, with a bandwidth of 0.14 nm applied for water density estimation.

Chloride conduction was measured in 200 ns simulations for each backbone-restrained receptor structure at 500 mM NaCl concentration and in the presence of a + 500 mV transmembrane potential difference, with positive potential on the cytoplasmic side. This was applied by imposing an external, uniform electric field in the membrane's normal direction.

The cryo-EM ivermectin binding poses were evaluated with gnina-torch (v1.6), a PyTorch implementation of the GNINA scoring function[97–99]. Pre-trained convolutional neural network models were used for protein-ligand scoring. The scores in Fig. 6 were obtained by the pre-trained default model ensemble from GNINA[99], which return an affinity (pK) and a variance.

Ivermectin was parameterized with the CHARMM General Force Field (CgenFF) and the Ligand Modeler of the CHARMM-GUI[100] with GROMACS (2021). After careful equilibration of the simulation system with GlyR and ivermectin based on the crystal binding pose, the stability of ivermectin was analyzed at different subunit interfaces in simulations with protein backbone restraints and in simulations without any restraints. The RMSD with respect to the cryo-EM binding pose was calculated with an in-house script using MDAnalysis (v2.3.0)[101,102].

## Reporting summary

Further information on research design is available in the Nature Portfolio Reporting Summary linked to this article.

## Data availability

All relevant data are available from the corresponding author upon request. The cryo-EM maps have been deposited in the Electron Microscopy Data Bank (EMDB) under accession codes EMD-26130 (αβGlyR-Stry), EMD-26141 (αβGlyR-Gly) and EMD-29019 (αβGlyR-Gly-Ivm). Coordinates have been deposited in the RCSB Protein Data Bank (PDB) under accession codes 7TU9 (αβGlyR-Stry), 7TVI (αβGlyR-Gly) and 8FE1 (αβGlyR-Gly-Ivm). Source data are provided in this paper.

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

## Acknowledgements

We would like to thank members of the Departments of Physiology and Biophysics and Pharmacology at Case Western Reserve University, specifically Dr. Tingwei Mu and his lab, for their help in setting up autopatch electrophysiology experiments, Dr. Ashutosh Prince and Dr. Rajesh Ramachandran for their help with confocal imaging, Dr. Xinghong Dai for tips on GO grid preparation, Dr. Wei Huang for providing an AlphaFold2 multimeric model, Dr. Anirban Mukherjee (Refeyn) for helping with Mass Photometry data collection and analysis, and members of the Dr. Sudha Chakrapani lab for their support in the lab and in preparing the manuscript. We would like to thank Dr. Kunpeng Li and Dr. Kyle Whiddon for their help with imaging at the CWRU cryo-EM facility. EG acknowledges training and cryo-EM access from the National Center for CryoEM Access and Training (NCCAT) and the Simons Electron Microscopy Center located at the New York Structural Biology Center, supported by the NIH Common Fund Transformative High-Resolution Cryo-Electron Microscopy program (U24 GM129539,) and by grants from the Simons Foundation (SF349247). S.L.I. was supported by the National Resource for Automated Molecular Microscopy, funded by NIH National Institute of General Medical Sciences (GM103310) and the Simons Foundation (SF349247). This work was supported by the National Institutes of Health grants R35GM134896 and Cryo-EM supplements R35GM134896-2S1 to S.C., F32GM142233 to E.G., AHA post-doctoral Fellowship 20POST35210394 to A.K. D.S is funded by the UKRI-BBSRC Interdisciplinary Bio-science Doctoral Training Partnership (BB/M011224/1). P.C.B. acknowledges support from BBSRC (BB/S001247/1).

## Author contributions

S.C. and E.G. designed and planned the project. E.G. optimized expression and purification of heteromeric GlyR, building off past work and assistance from A.K. E.G. and E.K. optimized expression of Domain E of gephyrin. E.G. and E.K. conducted whole-cell auto-patch electrophysiology experiments in HEK 293 T cells. E.K. analyzed electrophysiology data. E.G. collected and processed cryo-EM data and built atomic models with the help of A.K. S.I. provided guidance in processing cryo-EM data. D.S. designed and performed the computational analysis with guidance from P.B. S.C. supervised all aspects of the project. S.C. and E.G. prepared the manuscript text and figures with inputs from all the authors. All authors reviewed the final paper.

## Competing interests

The authors declare no competing interests.
