## [Peer Review File · Nature Communications]

Conformational transitions and allosteric modulation in a heteromeric glycine receptorREVIEWER COMMENTS

Reviewer #1 (Remarks to the Author):

Glycine receptor mediates fast inhibitory neurotransmission in the central nervous system. Gibbs et al here presents a series of cryo-EM structures of zebrafish $\alpha 1\beta$ glycine receptor, bound with strychnine, glycine and glycine/ivermectin. These new cryo-EM structures confirm the previously observed 4:1 stoichiometry in the formation of $\alpha:\beta$ hetero-pentameric complex, and reveal several new structural features in the N-terminal and ICD regions of glycine receptor. The authors propose that the N-terminal region of β subunit might be important in the assembly of 4:1 $\alpha:\beta$ complex. Overall, the structural works are interesting. By revealing the structures of different $\alpha 1\beta$ glycine receptor complex in the closed, desensitized and partially open states, this work advances our understanding the gating mechanism of glycine receptor. However, there are a few major issues need to be addressed before this work could be published. My specific points are:

(1) The structures of α and β subunits are highly similar. In the previous cryo-EM works, tag that is fused with β subunit or FAB that can specifically bind to α subunit were applied to make the α and β subunits more distinguishable. However, in this work, no tag or FAB was added to the zebrafish $\alpha 1\beta$ glycine receptor sample. The authors claim that the homo-pentameric and hetero-pentameric particles could be separated simply by 3D classification. I wonder the certainty and reliability of 3D classification results. The density figures in Fig S6 showing the subunit specific features of α and β subunits are poorly presented and are not convincing at all. In Fig S6A-C, both of the cryo-EM densities of α and β subunits are displayed at very low contour levels. This suggests that the regions where α and β subunits have distinct structural features were resolved at poor quality in the cryo-EM map. Therefore, it is possible that the cryo-EM maps of 4:1 α and β complex was reconstructed from compositionally distinct particles. To allow better separation of hetero-pentameric particles from the homo-pentameric particles as well prevent misalignment, ideally the authors need to add the FAB that could bind either α or β subunits to the glycine receptor sample. Given the high sequence similarities, the FAB developed for the α subunit of mammalian glycine receptor may bind to the α subunit of zebrafish glycine receptor equally well. In addition, it is very important that the modelling should be supported by the clear cryo-EM densities of the sidechains of those residues that are unique in either α or β subunits. The densities figure for showing the unique glycosylation sites between α or β subunits need to be improved significantly. They should be shown side-by-side in two different panel and with different orientations, and cryo-EM densities should be displayed at reasonable contour levels. The M2 helix is also quite different between α and β subunits. The model of M2s of α and β subunits together with cryo-EM density should be shown side-by-side, with those subunit specific residues highlighted. The sigma level for displaying the cryo-EM density should be indicated for each density figure.

(2) The cryo-EM density for the ICD region is resolved at very low resolution. No secondary structural features could be observed. The low resolution reconstruction could be also resulted from the mixture of homo- and hetero-pentameric particles and/or inaccurate particles alignment. Therefore, the segmentation of ICD density shown in Fig 6 is very questionable. Resolving the ICD region is one of the major novelties in this work. The authors need to improve the density for this region. They could collect more data, perform more extensive global/local 3D classification with symmetry expansion, and use FAB to facilitate the 3D classification and improve the accuracy of particle alignment.

(3) The author claims that ivermectin can only bind to α/β interface. This contradicts with all the previous structures showing that ivermectin can strongly bind to α/α interface. This is very surprising, as the key residues for ivermectin binding between α/β and α/α interfaces are quite similar. Is it possible that the amount of ivermectin in the cryo-EM sample is insufficient for saturating the receptor binding? To confirm that ivermectin can only bind to α/β interface in the 4:1 $\alpha:\beta$ hetero-pentameric complex, the author could perform in vitro binding experiment to show that the binding behavior of ivermectin to homo- and hetero-pentameric complexes are different.

(4) The densities of Stry and Gly should be shown together with the surrounding side-chain densities and in different orientations.

Reviewer #2 (Remarks to the Author):

The manuscript of Gibbs et al reports on three new structures of the full-length heteromeric glycine receptor (GlyR) $\alpha 1\beta$ from Zebrafish. These structures were solved by cryo-EM in the presence of antagonist (strychnine), agonist (glycine), or agonist with a positive allosteric modulator (glycine/ivermectin) and without exogenous fiducial markers, i.e., in the absence of protein alterations and/or antibodies. As such, this new set of structures could be useful to explore the conformational dynamics of native glycine receptors on gating as well as their allosteric modulation. These new structures: i. confirm the surprising 4 α :1 β stoichiometry observed in recent work (Yu Neuron 2021, Zhu Nature 2021); ii. illuminate previously unobserved structural features of the ECD and ICD in heteromeric GlyRs; and iii. highlight distinct and ligation-dependent ion-pore conformations with varying degrees of asymmetry. When augmented by Molecular Dynamics (MD) simulations, the new structures show that none of them is representative of the physiologically open state. Based on the conformation of the ion pore in cryo-EM (i.e., occluded at -2'), the in-silico electrophysiology results (i.e., non-conducting channels), and the comparison with previous glycine-bound desensitized states, it is concluded that the glycine-bound structure represents a desensitized receptor, whereas no functional annotation is given for the strychnine-bound or the IVM-bound structures. Last, the topographical location of the IVM-binding site is (tentatively) discussed as being in the TMD at the α/β interface only but no 3D model of it was provided, such that IVM binding remains unsolved.

Overall, the results presented in this manuscript seem to be useful and move forward (at least to some extent) our structural knowledge of synaptic receptors. However, I am not convinced that the level of novelty presented in this manuscript, particularly in relation to the recent structural work on heteromeric GlyRs, is so significant that justifies publication to a forum like Nature Communications. For example, it is unclear how many residues are resolved in the new structures that were previously unseen and how critical they are for function. In addition, several observations made in the manuscript appear intriguing but inconclusive, so that much expectation produced in the Title/Abstract/Introduction remains unmet. To mention a few: the characterization of the IVM-binding site with atomic resolution; the illumination of the conformation of the full ICD or part of it (i.e. the helical bundle formed by the post-M3 region); the functional annotation of structures in particular the one with IVM bound; their stability in a native-like lipid environment by unbiased MD, which was not accessed; an analysis of the global conformational changes of the receptor in terms of previously established collective variables like quaternary twisting and blooming (Cecchini Neuropharmacology 2015), which are commonly used as reaction coordinates to characterize receptor activation/deactivation (Calimet PNAS 2013, Lev PNAS 2017, Bergh eLife 2021), are all missing. Last, since no structure of the open state was obtained, no model of IVM-binding is proposed, and no functional interpretation of the global changes between the cryo-EM structures is provided, the title of the manuscript appears as most inappropriate.

Based on the above, I am afraid I cannot support publication of this manuscript in Nature Communications.

Reviewer #3 (Remarks to the Author):

This paper by Gibbs et al presents the cryo-EM structures of the full-length zebrafish $\alpha 1\beta$ GlyR in three distinct conformations, in the presence of strychnine (annotated as resting state), glycine (desensitized) and glycine plus ivermectin (semi open or desensitized). In addition, MD simulations are performed to document the conductive/non-conductive properties of the channel in the different conformations. The receptor is expressed together with the anchoring protein gephyrin E chain that is known to bind to the cytoplasmic loop predicted to be unstructured. This work follows two previous articles on a similar topic, the structure of the native pig receptor, and the structure of recombinant human $\alpha 2\beta$ receptor in three conformations, in complex with strychnine (closed) and glycine (desensitized and semi open).

The text and the figures are clear, the cryo-EM structures are well presented and discussed in relation

to previous structural work and the relevant literature. The new structures recapitulate previous finding, in particular the one beta to four alpha stoichiometry, as well as the overall conformation of the receptor in resting and desensitized conformations. In this sense, the work does not bring original information on the system, especially since the previous structures by Yu et al were performed on human receptors that are more relevant in physiopathology, although they were performed on receptors truncated from the cytoplasmic domain. This limitation of the work is well discussed in the manuscript, but authors highlight specific features of the structures that were neither observed. Among them one can distinguish:

1/ identification of a density that could correspond to a portion of ivermectin at the α/β interface, binding in a different way to the well characterized binding locus on homomeric receptors. This suggests a quite different mechanism of action of ivermectin on heteromers versus homomers. However, the density is only partial, and a convincing demonstration that it indeed corresponds to the potentiation binding site could be performed by mutagenesis/electrophysiology. This would have interesting implication in drug-design purpose.

2/ previously unresolved portion of the N-terminus of the beta subunits, as well as of the C-terminus of the alpha subunit are resolved in the structure, which is of interest.

3/ Perhaps the most interesting finding is the visualization of a portion of the ICD, allowed thanks to the use of full-length subunits and possibly the presence of gephyrin. However, the local resolution is not sufficient and the signal partial, and authors were able only to partially resolve extensions of the M3 helix for each of the subunits. A speculative partial model is constructed using alphafold. According to these data, the ICD appears strongly asymmetric, away from the beta subunit. Although the structure is not at high resolution at this level, the work provides a first glimpse of the ICD which is important in synaptic anchoring and allosteric regulation by phosphorylation, G proteins and ethanol. An additional supplementary figure, showing the alphafold model including both the TMD and ICD, would be valuable for the reader to better visualized the orientation of the ICD helices in relation to the TMD helices.

In conclusion, although somewhat lacking novelty concerning the overall structure of heteromeric GlyR, the work brings important new information on domains and binding sites that were out of reach in previous studies. The work is nicely presented and will be of interest to a large community of scientists. Better characterization of the putative ivermectin binding site and/or demonstration that it mediate potentiation would be valuable.

Response to Reviewer Comments

Thank you to the reviewers for their thoughtful and well-articulated comments. We noted that while they found the presented work interesting, they also raised several points of concern. Overall, we agree with these valid concerns and have addressed them with additional experiments included in the revision. A summary of new data included in the revision:

- To address the issue of partial occupancy of ivermectin in the previous data set, we explored additional conditions for sample preparation. Particularly, incubating the sample with ivermectin (20 μ M) prior to adding glycine (1 mM). While the ligand concentrations were similar to the previous data set (where the two ligands were co-incubated), in the new Cryo-EM data set (2.98 Å), the map shows clear ivermectin density at each interface. Quite remarkably, the ivermectin binding pose at the β - α interface is distinct from the α - α interfaces (more details below).
- We carried out MD simulation of the new $\alpha\beta$ GlyR-Gly-Ivm structure to assess the conductance state of the channel in this conformation, validate the stability of the ivermectin binding poses, and map the interaction fingerprint of ivermectin at the different interfaces.
- We carried out mass photometry to assess sample heterogeneity.
- We have updated the figures to better illustrate subunit-specific differences (including a movie to highlight this) and made substantial changes to the discussion (with additional references for previous MD work).

Response to Reviewer 1:

Glycine receptor mediates fast inhibitory neurotransmission in the central nervous system. Gibbs et. al here presents a series of cryo-EM structures of zebrafish $\alpha 1\beta$ glycine receptor, bound with strychnine, glycine and glycine/ivermectin. These new cryo-EM structures confirm the previously observed 4:1 stoichiometry in the formation of $\alpha:\beta$ hetero-pentameric complex, and reveal several new structural features in the N-terminal and ICD regions of glycine receptor. The authors propose that the N-terminal region of β subunit might be important in the assembly of 4:1 $\alpha:\beta$ complex. Overall, the structural works are interesting. By revealing the structures of different $\alpha 1\beta$ glycine receptor complex in the closed, desensitized and partially open states, this work advances our understanding the gating mechanism of glycine receptor.

We appreciate the reviewer's time and effort to understand our work and its context within the field. We are grateful for the detailed comments and concerns, which have pushed us to improve our manuscript.

However, there are a few major issues need to be addressed before this work could be published. My specific points are:

1) The structures of α and β subunits are highly similar. In the previous cryo-EM works, tag that is fused with β subunit or FAB that can specifically bind to α subunit were applied to make the α and β subunits more distinguishable. However, in this work, no tag or FAB was added to the zebrafish $\alpha 1\beta$ glycine receptor sample. The authors claim that the homo-pentameric and hetero-

pentameric particles could be separated simply by 3D classification. I wonder the certainty and reliability of 3D classification results. The density figures in Fig S6 showing the subunit specific features of α and β subunits are poorly presented and are not convincing at all. In Fig S6A-C, both of the cryo-EM densities of α and β subunits are displayed at very low contour levels. This suggests that the regions where α and β subunits have distinct structural features were resolved at poor quality in the cryo-EM map. Therefore, it is possible that the cryo-EM maps of 4:1 α and β complex was reconstructed from compositionally distinct particles. To allow better separation of heteropentameric particles from the homo-pentameric particles as well prevent misalignment, ideally the authors need to add the FAB that could bind either α or β subunits to the glycine receptor sample. Given the high sequence similarities, the FAB developed for the α subunit of mammalian glycine receptor may bind to the α subunit of zebrafish glycine receptor equally well. In addition, it is very important that the modelling should be supported by the clear cryo-EM densities of the sidechains of those residues that are unique in either α or β subunits. The densities figure for showing the unique glycosylation sites between α or β subunits need to be improved significantly. They should be shown side-by-side in two different panel and with different orientations, and cryo-EM densities should be displayed at reasonable contour levels. The M2 helix is also quite different between α and β subunits. The model of M2s of α and β subunits together with cryo-EM density should be shown side-by-side, with those subunit specific residues highlighted.

The reviewer raises two potential reasons that the cryo-EM density may not be representative of heteromeric GlyR, sample heterogeneity arising from multiple stoichiometries, and particle misalignment during cryo-EM image processing.

Our protocol for heteromeric $\alpha 1\beta$ -GlyR-GephE with two affinity purification steps consistently yields a near monodisperse peak with bands for all three proteins in western-blot analysis. The elution peak from size exclusion chromatography is 0.75 ml left-shifted compared to homomeric $\alpha 1$ -GlyR preparations under similar conditions. There was also no evidence of true homomeric channels in our cryo-EM processing, but rather the early stages of refinement produced pseudo-symmetric C5 classes that had mixed features of α and β subunits.

To better assess possible particle heterogeneity in our sample, we performed mass photometry experiments, which are shown below. Mass photometry is an optical diffraction based method that counts events by induced contrast, which is linearly correlated with protein mass. Count distributions are Poissonian and the peak gives a mass estimate within 2% for protein standards. Our sample yielded peaks at 44 kDa, 86 kDa, 396 kDa, 804 kDa and very a small peak at 1121 kDa. The 44 kDa peak is slightly smaller than a detergent micelle, but was similar to what was observed in a sample containing only 1 mM DDM. Per the manufacturer, molecular weights below 60 kDa are less reliable. The predicted molecular weights for each gene, including affinity tags, are 49.8, 63.2 and 49.6 kDa for α GlyR, β GlyR and Geph-E respectively. The predicted molecular weight for a 4 α :1 β complex bound to a Geph-E dimer is 361.6 kDa. Adding 44 kDa for the detergent micelle gives 405.5 kDa. Given the uncertainty of the detergent mass surrounding GlyR, this is reasonably close to the expected measured mass of 396 kDa. A homomeric assembly of α GlyR would be expected to have molecular weight of 293.1 kDa, hence the data reasonably excludes a sizable population of homomeric α GlyR. Though not conclusive, the most likely

interpretation is that of a $4\alpha:1\beta$ complex, consistent with gel filtration and this manuscript's and other's cryo-EM results. Regarding the larger mass species in the data, coordinated binding of two receptors to a single Geph-E dimer gives a predicted mass of 668.0 kDa. Hence the peak at 804 kDa likely corresponds to non-specific aggregates of the 396 kDa species. Hence, we conclude that our expression system produces heteromeric GlyR that is expressed at a fixed stoichiometry, a finding consistent with past structural and biochemical studies.

The reviewer also correctly notes the computational challenges with particle alignment given the inherent similarities between α and β subunits. Along the same line as our work, a recent report on heteromeric nAChRs showed that similar structural differences (glycan locations, the length of the M4 C-termini) were sufficient to distinguish between the homologous subunits without external fiduciary markers (Zarkadas et al Neuron 2022). This challenge is a worthwhile pursuit because binding to the Fab or truncating the protein has consequences of its own. For instance, Fab binding in the vicinity of the neurotransmitter binding pocket limits transitions to certain other functional states.

In this work, the differences between $\alpha 1$ and β subunits are subtle but clear and consistent across the structures in our manuscript, previously published heteromeric GlyR structures, AlphaFold predictions and previously published biochemical data. Using the reviewer's suggestions, we have

updated what is now Supplemental Figure 7 and included Supplementary Movie 1 to better support our claim to resolve separate subunits. The movie compares $\alpha 1\beta$ GlyR-Gly-IVM to our previously published map of homomeric $\alpha 1$ GlyR-Gly-IVM of comparable resolution (2.98Å vs 3.07 Å). We have also shared the new maps if that is helpful to the reviewer.

To summarize the key notable differences between α and β subunits are:

- Length of M4 helix- There are 10 additional amino acids at the c-terminal end of M4 in the α subunit constituting the post-M4 region, this region is missing in the β subunit.

- Glycans- Density for subunit-specific glycans is observed at each subunit (N62 in α GlyR and N241 in β GlyR)

- Additional density is present at the N-terminal end of β GlyR ECD.

- The ivermectin binding pose – There is a distinct ivermectin binding pose at the β - α interface in comparison to the α - α interfaces. The differences at the ivermectin binding site in the map are compelling evidence of our ability to resolve meaningful differences between subunits. Previous studies have highlighted the importance of side-chain volume at position A312 of α 1GlyR and the equivalent position in the glutamate-gated chloride channel (GluCl).

Particularly, a larger residue (phenylalanine) eliminates ivermectin agonist activity in homomeric GlyR α 1 (Lynagh 2011). The β subunit has an isoleucine at this position and accordingly we see both a larger side-chain density and a different conformation of the ivermectin density at this interface compared to others (Fig 6, S. Fig 7). There are also differences in the charged and polar residues at each interface that form hydrogen bonds between ivermectin and side chains of M1 and M2. These are discussed more fully in the revised text.

As suggested we have also updated Figure 2 to include the sequence of both subunits.

Though each example is subtle on its own, the agreement between various heteromeric features convinces us that the map is indeed representative of heteromeric GlyR.

The sigma level for displaying the cryo-EM density should be indicated for each density figure.

The sigma level is now reported in each of the figure legends.

(2) The cryo-EM density for the ICD region is resolved at very low resolution. No secondary structural features could be observed. The low resolution reconstruction could be also resulted from the mixture of homo- and hetero-pentameric particles and/or inaccurate particles alignment. Therefore, the segmentation of ICD density shown in Fig 6 is very questionable. Resolving the ICD region is one of the major novelties in this work. The authors need to improve the density for this region. They could collect more data, perform more extensive global/local 3D classification with symmetry expansion, and use FAB to facilitate the 3D classification and improve the accuracy of particle alignment.

The ICD has been of great interest to the community given its critical role in channel function, trafficking and synaptic regulation. The domain E of gephyrin was included as a strategy to improve the signal from this region. In spite of this complex, the cryo-EM density is indeed weak within our ICD reconstruction. However, we included it since it is the first view of this domain which has been intractable in the previously reported reconstructions. The asymmetric positioning of the domain is interesting and may have mechanistic implications.

The low-resolution ICD reconstruction is not necessarily indicative of mixed particles or misalignment as the ICD is also weakly resolved in the Fab-bound structure of Zhu et al. (See comment/response #2 from reviewer 1 in Peer Review File). Collecting more data could slightly improve the intracellular domain, but given the already large number of movies (14,270) it would require significant time with likely marginal gains. Symmetry expansion is only appropriate for compositionally identical subunits with different orientations, as would be the case for α homomeric channel. Symmetry relaxation is an appropriate technique for heterogeneous channels. As such, we collaborated with Dr. Serban Ilca (Postdoctoral Scholar at NYSBC) one of the original developers of the technique (also a co-author in this manuscript). The present map is the result of our combined efforts.

Upon reflection, we do agree that the segmentation and electrostatics representation may be too preliminary to be included in the manuscript. We have therefore replaced these figures and have simplified our conclusions in the updated text.

(3) The author claims that ivermectin can only bind to α/β interface. This contradicts with all the previous structures showing that ivermectin can strongly bind to α/α interface. This is very surprising, as the key residues for ivermectin binding between α/β and α/α interfaces are quite similar. Is it possible that the amount of ivermectin in the cryo-EM sample is insufficient for saturating the receptor binding? To confirm that ivermectin can only bind to α/β interface in the 4:1 $\alpha:\beta$ hetero-pentameric complex, the author could perform in vitro binding experiment to show that the binding behavior of ivermectin to homo- and hetero-pentameric complexes are different.

The new $\alpha\beta$ GlyR-Gly-Ivm structure shows ivermectin binding at all five subunit interfaces. There are key differences in the EM density between $\alpha-\alpha$, $\alpha-\beta$ and $\beta-\alpha$ interfaces. This is consistent with previous mutagenesis studies of the ivermectin binding site and provides a novel perspective of how ivermectin binds and exerts effects on heteromeric GlyR. For more details, please review the section on ivermectin binding in the updated manuscript.

The lack of ivermectin at the $\alpha-\alpha$ interface in the previous data set was indeed surprising and we did not intend to convey any strong claims. The possibility of conformational heterogeneity at sub-saturating ivermectin concentrations is interesting and may be explored in future work.

(4) The densities of Stry and Gly should be shown together with the surrounding side-chain densities and in different orientations.

Fig 5, which shows the strychnine and glycine binding site, has been updated with multiple orientations and side chain densities as the reviewer suggested.

Response to Reviewer 2:

The manuscript of Gibbs et al reports on three new structures of the full-length heteromeric glycine receptor (GlyR) $\alpha 1\beta$ from Zebrafish. These structures were solved by cryo-EM in the presence of antagonist (strychnine), agonist (glycine), or agonist with a positive allosteric modulator (glycine/ivermectin) and without exogenous fiduciary markers, i.e., in the absence of protein alterations and/or antibodies. As such, this new set of structure could be useful to explore the conformational dynamics of native glycine receptors on gating as well as their allosteric modulation. These new structures: i. confirm the surprising 4 α :1 β stoichiometry observed in recent work (Yu Neuron 2021, Zhu Nature 2021); ii. illuminate previously unobserved structural features of the ECD and ICD in heteromeric GlyRs; and iii. highlight distinct and ligation-dependent ion-pore conformations with varying degrees of asymmetry. When augmented by Molecular Dynamics (MD) simulations, the new structures show that none of them is representative of the physiologically open state. Based on the conformation of the ion pore in cryo-EM (i.e., occluded at -2'), the in-silico electrophysiology results (i.e., non-conducting channels), and the comparison with previous glycine-bound desensitized states, it is concluded that the glycine-bound structure represents a desensitized receptor, whereas no functional annotation is given for the strychnine-bound or the IVM-bound structures. Last, the topographical location of the IVM-binding site is (tentatively) discussed as being in the TMD at the α/β interface only but no 3D model of it was provided, such that IVM binding remains unsolved.

We would like to thank the reviewer for taking the time to review our work. We appreciate their comments that highlight the significance of the new findings but also point out the gaps in the manuscript. Our revision addresses these weaknesses through new data provided and with substantial changes to the discussion.

Overall, the results presented in this manuscript seem to be useful and move forward (at least to some extent) our structural knowledge of synaptic receptors. However, I am not convinced that the level of novelty presented in this manuscript, particularly in relation to the recent structural work on heteromeric GlyRs, is so significant that justifies publication to a forum like Nature Communications. For example, it is unclear how many residues are resolved in the new structures that were previously unseen and how critical they are for function.

Our revised manuscript has several novel contributions that are discussed in specific examples below. Our manuscript is also valuable as it verifies the controversial results of past structural work. The recent heteromeric structures overturned the accepted paradigm of glycine receptor stoichiometry. In their work with native receptors, Zhu et al. express that their purification strategy may bias the stoichiometry towards the $4\alpha:1\beta$ complex. Our work presents an alternate purification strategy and expression system that produces the same subunit stoichiometry. This strengthens the argument that heteromeric GlyR fundamentally assembles a predominant, if not exclusive, $4\alpha:1\beta$ complex. In addition to providing new details of conformational changes associated with activation, desensitization, and potentiation in a full-length heteromeric-GlyRs (in complex with gephyrin-E), our work highlights the low-resolution features of the ICD that were not discussed in the past work. While not of high resolution, these provide a useful framework for future research. There is considerable interest of even a “potato density” of the ICD (Comment 2 of reviewer 1 in Zhu et al. 2021). The role of the highly conserved β subunit-specific N-terminal sequence is also fascinating precisely because it is unknown.

We have tried to clarify these points in the discussion.

In addition, several observations made in the manuscript appear intriguing but inconclusive, so that much expectation produced in the Title/Abstract/Introduction remains unmet. To mention a few: the characterization of the IVM-binding site with atomic resolution;

The new $\alpha\beta$ GlyR-Gly-Ivm structure offers unique high-resolution details of ivermectin binding at subunit specific interfaces and clearer evidence that the addition of ivermectin results in a distinct desensitized state from glycine alone.

the illumination of the conformation of the full ICD or part of it (i.e. the helical bundle formed by the post-M3 region);

As discussed above, despite our best efforts, our resolution of the ICD density is limited. The updated manuscript gives more conservative conclusions.

the functional annotation of structures in particular the one with IVM bound;

Per suggestion, we have updated the manuscript with clear annotations of each structural state. Based on the conformation (and compared to the previous α 1GlyR-apo), we refer to the $\alpha\beta$ -GlyR-Stry as the closed, $\alpha\beta$ -GlyR-Gly and $\alpha\beta$ -GlyR-Gly-IVM as desensitized states.

their stability in a native-like lipid environment by unbiased MD, which was not accessed;

We have performed simulations to assess the stability of ivermectin at the different interface sites (see new Figure 6C). The dynamics of these models is consistent with reasonable binding modes.

an analysis of the global conformational changes of the receptor in terms of previously established collective variables like quaternary twisting and blooming (Cecchini Neuropharmacology 2015), which are commonly used as reaction coordinates to characterize receptor activation/deactivation (Calimet PNAS 2013, Lev PNAS 2017, Bergh eLife 2021), are all missing.

The global conformational changes described in the manuscript are comparisons between cryo-EM structures. A limitation is that we cannot accurately state whether it is the ECD or TMD that rotates, only that their relative positions have changed. However, the observed relative rotation is consistent with the twisting motion and the updated manuscript includes the suggested reference that describes this motion. Interestingly the blooming motion, or outward displacement of the ECD is not observed between the various functional states in this manuscript. Rather, the center of gravity of an individual subunit ECD is fixed, but there is a rigid rotation of the subunit as described in Figure 4 that causes the top of the ECD to expand inward and the bottom outward. There is an outward displacement of the transmembrane helix bundle. The revised manuscript discusses similarities and differences in global conformational changes of this work compared to those described in the provided references.

Last, since no structure of the open state was obtained, no model of IVM-binding is proposed, and no functional interpretation of the global changes between the cryo-EM structures is provided, the title of the manuscript appears as most inappropriate.

While the presented work does not include a model of the open channel, the manuscript provides details of global conformational changes associated with heteromeric GlyR activation and desensitization, and in the revision high-resolution description of ivermectin binding and potentiation. We note that the reviewer is concerned about the original title “*Conformational Dynamics of Heteromeric Glycine receptors during Gating and Allosteric Modulation*”. We have updated the title to “*Conformational transitions in a heteromeric glycine receptor associated with antagonism, agonism, and positive allosteric modulation*”

Response to Reviewer 3:

This paper by Gibbs et al presents the cryo-EM structures of the full-length zebrafish alpha1-beta GlyR in three distinct conformations, in the presence of strychnine (annotated as resting state), glycine (desensitized) and glycine plus ivermectin (semi open or desensitized). In addition, MD simulations are performed to document the conductive/non-conductive properties of the channel in the different conformations. The receptor is expressed together with the anchoring protein gephyrin E chain that is known to bind to the cytoplasmic loop predicted to be unstructured. This

work follows two previous articles on a similar topic, the structure of the native pig receptor, and the structure of recombinant human alpha2 beta receptor in three conformations, in complex with strychnine (closed) and glycine (desensitized and semi open).

The text and the figures are clear, the cryo-EM structures are well presented and discussed in relation to previous structural work and the relevant literature. The new structures recapitulate previous finding, in particular the one beta to four alpha stoichiometry, as well as the overall conformation of the receptor in resting and desensitized conformations. In this sense, the work does not bring original information on the system, especially since the previous structures by Yu et al were performed on human receptors that are more relevant in physiopathology, although they were performed on receptors truncated from the cytoplasmic domain.

We would like to thank the reviewer for reviewing the manuscript. They correctly characterize the work and its relation to other recent work.

This limitation of the work is well discussed in the manuscript, but authors highlight specific features of the structures that were neither observed. Among them one can distinguish: 1/ identification of a density that could correspond to a portion of ivermectin at the $\alpha A/\beta$ interface, binding in a different way to the well characterized binding locus on homomeric receptors. This suggests a quite different mechanism of action of ivermectin on heteromers versus homomers. However, the density is only partial, and a convincing demonstration that it indeed corresponds to the potentiation binding site could be performed by mutagenesis/electrophysiology. This would have interesting implication in drug-design purpose.

As previously stated, we have replaced the ivermectin structure with partial occupancy for a new one that shows complete occupancy. In this new structure, there are notable differences in ivermectin binding at distinct subunit interfaces. Like the reviewer, we are excited about possible effects of these differences on partial occupancy and will explore them in future work. However, for immediate clarity we only present the new, full occupancy, structure in this manuscript.

2/ previously unresolved portion of the N-terminus of the beta subunits, as well as of the C-terminus of the alpha subunit are resolved in the structure, which is of interest.

Indeed. We have therefore retained this information in the revised manuscript but have toned-down our interpretation in the discussion.

3/ Perhaps the most interesting finding is the visualization of a portion of the ICD, allowed thanks to the use of full-length subunits and possibly the presence of gephyrin. However, the local resolution is not sufficient and the signal partial, and authors were able only to partially resolve extensions of the M3 helix for each of the subunits. A speculative partial model is constructed using alphafold. According to these data, the ICD appears strongly asymmetric, away from the beta subunit. Although the structure is not at high resolution at this level, the work provides a first glimpse of the ICD which is important in synaptic anchoring and allosteric regulation by phosphorylation, G proteins and ethanol. An additional supplementary figure, showing the alphafold model including both the TMD and ICD, would be valuable for the reader to better visualized the orientation of the ICD helices in relation to the TMD helices.

AlphaFold predictions for each subunit have been included in Figure 7. Additionally, we looked at the prediction for the multimer. However, there was no structural difference was noted between the ICD portions of each subunit, and no helix bundle as our density would suggest. Considering that our complex includes Geph-E (which may induce structural changes to the ICD) and given the low predictive power of AlphaFold beyond the initial post-M3 and pre-M4 segments, we are hesitant to over-interpret these results.

In conclusion, although somewhat lacking novelty concerning the overall structure of heteromeric GlyR, the work brings important new information on domains and binding sites that were out of reach in previous studies. The work is nicely presented and will be of interest to a large community of scientists. Better characterization of the putative ivermectin binding site and/or demonstration that it mediate potentiation would be valuable.

Thank you for the thoughtful review.

REVIEWER COMMENTS

Reviewer #1 (Remarks to the Author):

The authors have addressed most of my concerns raised from the first round of review, but the mass photometry results used for supporting the sample homogeneity is not convincing, due to the lack of control experiments. The authors also need to measure the molecule weight of homo-pentamer and hetero-pentamer without Geph-E bound, and confirm that these two complexes indeed have smaller molecule weight than the hetero-pentamer with Geph-E bound. These results could be shown in the manuscript to support the sample quality.

Reviewer #2 (Remarks to the Author):

In the revised version, the authors have addressed part of my concern with new data and changes in the discussion. The revised manuscript includes a refined structure of heteromeric GlyR in complex with IVM, which provides high-resolution details of IVM binding at different subunit-subunit interfaces, along with a revised discussion that focuses on differences and similarities with previous structural work on both GlyR (heteromeric and homomeric) and homologous channels. The changes introduced, particularly the atomistic model of IVM binding, respond to some of my criticism. However, no atomistic model of the ICD could be provided, and little mechanistic understanding of receptor's function and allosteric modulation emerges from this study. Despite the potential interest/relevance of these heteromeric structures, I regret I cannot recommend the manuscript for publication to Nature Communication.

Reviewer #3 (Remarks to the Author):

Authors have addressed my concerns

Response to Reviewer Comments

We once again thank all three reviewers for their feedback. We are delighted that the reviewers feel that we have addressed most of their concerns from the previous round of review. We have made the following changes to the manuscript based on their suggestion.

Reviewer #1

The authors have addressed most of my concerns raised from the first round of review, but the mass photometry results used for supporting the sample homogeneity is not convincing, due to the lack of control experiments. The authors also need to measure the molecule weight of homo-pentamer and hetero-pentamer without Geph-E bound, and confirm that these two complexes indeed have smaller molecule weight than the hetero-pentamer with Geph-E bound. These results could be shown in the manuscript to support the sample quality.

We agree with the Reviewer's point that it would be ideal to also directly measure the molecular weight of the homomeric GlyR and heteromeric GlyR (without Geph-E) to compare with the mass of heteromeric GlyR with Geph. In the interest of time, (since we do not have local access to mass photometry), we have now included gel filtration profiles of homomeric GlyR and heteromeric GlyR (with and without Geph-E) that shows that in complex with Geph-E there is a leftward shift in elution. Based on the Reviewer's suggestion we have included all this information in Supplemental Figure 1.

Reviewer #2:

In the revised version, the authors have addressed part of my concern with new data and changes in the discussion. The revised manuscript includes a refined structure of heteromeric GlyR in complex with IVM, which provides high-resolution details of IVM binding at different subunit-subunit interfaces, along with a revised discussion that focuses on differences and similarities with previous structural work on both GlyR (heteromeric and homomeric) and homologous channels. The changes introduced, particularly the atomistic model of IVM binding, respond to some of my criticism. However, no atomistic model of the ICD could be provided, and little mechanistic understanding of receptor's function and allosteric modulation emerges from this study. Despite the potential interest/relevance of these heteromeric structures, I regret I cannot recommend the manuscript for publication to Nature Communication.

We note that the Reviewer finds that the main concerns about Ivermectin binding site are now resolved. We acknowledge the lingering criticism from this Reviewer on the lack of high-resolution information on the ICD. We are continuing to pursue this area through additional strategies.

Reviewer #3

Authors have addressed my concerns

Thank you.